# GRAPH NEURAL NETWORKS ON SYMMETRIC POSITIVE DEFINITE MANIFOLD

## ABSTRACT

Geometric deep learning endows graph neural networks (GNNs) with some symmetry aesthetics from the inherent principles of the underlying graph structures. However, conventional modeling in Euclidean or hyperbolic geometry, often presupposes specific geometric properties for graphs, thereby neglecting the intricate actual structures. To address this limitation, this study generalizes the foundational components of GNNs to the Symmetric Positive Definite (SPD) manifold. This manifold theoretically endowed with a rich geometric structure that encompasses both Euclidean and hyperbolic projection subspaces. Motivated by this, we reconstruct GNNs with manifold-preserving linear transformation, neighborhood aggregation, non-linear activation, and multinomial logistic regression. In this framework, the Log-Cholesky metric is employed to derive the closed-form Fréchet mean representation for neighborhood aggregation, ensuring computational tractability in learning geometric embeddings. Experiments demonstrate that the SPDGNN can learn superior representations for grid and hierarchical structures, leading to significant performance improvements in subsequent classifications compared to the Euclidean and hyperbolic analogs.

## 1 INTRODUCTION

Geometric deep learning seeks to extend the application of deep learning techniques beyond Euclidean domains, encompassing graphs and manifolds (Bronstein et al., 2021). Graph neural networks (GNNs) represent a specific category of geometric deep learning models tailored for graph data. Capitalizing on the intrinsic geometric properties of graphs, these networks effectively encode extensive information concerning both node attributes and graph topology. Within diverse learning communities, the embeddings derived through GNNs have demonstrated noteworthy success across various graph-related tasks, including the link prediction (Chami et al., 2019), graph classification (Xu et al., 2018), and node classification (Kipf & Welling, 2017).

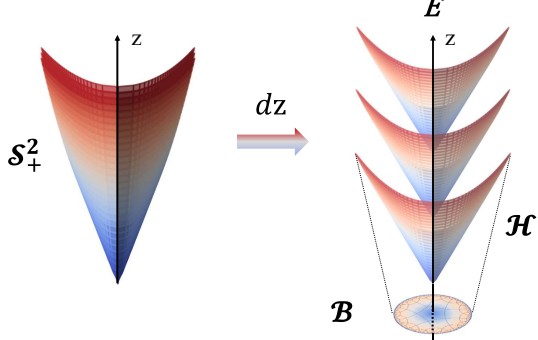

Figure 1: The SPD manifold $\mathcal{S}_+^2$ (w.r.t. Eqn. 1) could be approximately integrated by its manifold slice (hyperboloid $\mathcal{H}$) over Euclidean observation direction ($d\mathrm{z}$), and then be sequentially projected into the non-Euclidean Poincaré ball $\mathcal{B}$. From geometry perspective, SPD contains Euclidean and hyperbolic entailments.

Despite the success of Euclidean embeddings, recent research (Bronstein et al., 2021) has brought to light that many complex graph datasets manifest characteristics that surpass the purview of conventional Euclidean analysis. In such circumstances, the utilization of Euclidean space inevitably introduces geometric induction bias into the graph representation. Conversely, hyperbolic space has emerged as a viable alternative to Euclidean space, demonstrating efficacy in embedding tree structures with minimal distortion (Ganea et al., 2018; Chami et al., 2019; Zhang et al., 2021b). However, whether modeling in Euclidean, hyperbolic, or their Cartesian product spaces, it typically entails hypothesizing that the structural properties of the graph adhere to specific geometric preferences, thereby distorting the genuine graph structure.

To transcend the intrinsic geometric preferences imposed on graph data by Euclidean or hyperbolic embedding spaces, it is valuable to explore a more solid framework for geometric representations. The Symmetric Positive Definite (SPD) manifold, theoretically endowed with a rich geometric structure that encompasses both Euclidean and hyperbolic projection subspaces, is a promising avenue for investigation (Helgason, 1979). From a geometry decomposition perspective, SPD may be approximated by integrating Euclidean and hyperbolic geometric structures, as illustrated in Fig. 1. Embedding graphs into the SPD manifold offers significant advantages, as it allows for the concurrent modeling of hierarchical structures in hyperbolic subspaces and grid structures in Euclidean subspaces. This versatility overcomes the constraints inherent in exclusively relying on either hyperbolic or Euclidean spaces. Notably, the distinctive sub-manifold geometry enables the identification and differentiation of substructures within the graph (López et al., 2021a;b).

In this paper, we present a novel reconstruction of GNNs termed SPDGNN, which generalizes the fundamental components of GNNs onto the SPD manifold. Our innovative approach encompasses manifold-preserving linear transformation, neighborhood aggregation, non-linear activation, and multinomial logistic regression. Notably, the utilization of the Log-Cholesky metric (Lin, 2019) in this framework facilitates the derivation of the closed-form Fréchet mean representation for neighborhood aggregation, ensuring computational tractability in learning geometric embeddings. To evaluate the performance of SPDGNN, we conduct experiments on four publicly available real-world datasets for the task of semi-supervised node classification. The experimental results demonstrate the effectiveness of our proposed SPDGNN in acquiring enhanced representations for Euclidean and hyperbolic geometries.

## 2 GEOMETRY OF THE SPD MANIFOLD WITH LOG-CHOLESKY METRIC

**Riemannian geometry.** A manifold $\mathcal{M}$ with $n$ dimensions is a topological space that can be locally approximated by an $n$-dimensional real space $\mathbb{R}^n$ at any point $p \in \mathcal{M}$, known as the tangent space $\mathcal{T}_p\mathcal{M}$. A Riemannian manifold $\mathbb{L}$ is a differentiable manifold $\mathcal{M}$ endowed with a Riemannian metric tensor field $\hat{\mathfrak{g}}$, expressed as $\mathbb{L} = (\mathcal{M}, \hat{\mathfrak{g}})$. Given two points $u, v \in \mathcal{T}_p\mathcal{M}$, Riemannian metric $\hat{\mathfrak{g}}$ defines an inner product on the tangent space $\mathcal{T}_p\mathcal{M}$ such that $\hat{\mathfrak{g}}_p(u, v) := \langle u, v \rangle_p$. Let $\gamma : [0, 1] \to \mathcal{M}$ be a smooth parameterized curve on $\mathcal{M}$ with velocity vector at $t \in [0, 1]$ denoted as $\dot{\gamma}_t \in \mathcal{T}_{\gamma(t)}\mathcal{M}$, where $\| \cdot \|_{\gamma_t}$ denotes the Riemannian norm. The length of curve $\gamma$ is defined as $L_\gamma = \int_0^1 \|\dot{\gamma}_t\|_{\gamma_t} dt = \int_0^1 \sqrt{\hat{\mathfrak{g}}_{\gamma(t)}(\dot{\gamma}_t, \dot{\gamma}_t)} dt$. Any two points $p, q \in \mathcal{M}$ can be joined by a unique length-minimizing curve, called a geodesic $d_\mathcal{M}(p, q)$. Moreover, for each point $p \in \mathcal{M}$ and a tangent vector $v \in \mathcal{T}_p\mathcal{M}$, the Riemannian exponential map $\text{Exp}_p : \mathcal{T}_p\mathcal{M} \to \mathcal{M}$ maps the tangent vector $v$ onto the manifold $\mathcal{M}$. The Riemannian logarithmic map is the reverse map that maps the point $\text{Exp}_p(v)$ back to the tangent space at $p$ such that $\text{Log}_p(\text{Exp}_p(v)) = v$. Another pivotal operation is the parallel transport $P_{p \to q} : \mathcal{T}_p\mathcal{M} \to \mathcal{T}_q\mathcal{M}$, enabling the transportation of the tangent vectors from point $p$ to $q$ along the unique geodesic while preserving the metric tensor $\hat{\mathfrak{g}}$.

**SPD manifold.** A symmetric matrix is deemed positive definite if all its eigenvalues are positive. The space of $n \times n$ SPD matrices (Helgason, 1979) is denoted by

$$\mathcal{S}_+^n = \{\boldsymbol{A} \in \mathbb{R}^{n \times n} | \boldsymbol{A} = \boldsymbol{A}^\top, \boldsymbol{A} \succ 0\}, \tag{1}$$

where $\boldsymbol{A}^\top$ denotes the transpose of matrix $\boldsymbol{A}$, and $\boldsymbol{A} \succ 0$ indicates that all eigenvalues of $\boldsymbol{A}$ are positive. The space of SPD matrices $\mathcal{S}_+^n$ constitutes a convex smooth manifold in the Euclidean space $\mathbb{R}^{n \times (n+1)/2}$, and various inherited metrics further transform $\mathcal{S}_+^n$ into a Riemannian manifold (Pennec et al., 2006; Arsigny et al., 2007; Lin, 2019). The tangent space at any point $\boldsymbol{P} \in \mathcal{S}_+^n$ can be identified as the vector space of symmetric matrices $\mathcal{S}^n$, denoted as $\mathcal{T}_{\boldsymbol{P}}\mathcal{S}_+^n$. The standard matrix exponential and logarithm of matrix $\boldsymbol{P}$ are represented as $\exp(\boldsymbol{P})$ and $\log(\boldsymbol{P})$, respectively, serving as a diffeomorphism between manifolds $\mathcal{S}_+^n$ and $\mathcal{S}^n$.

**Log-Cholesky metric.** We formally introduce the concept of a Riemannian manifold $(\mathcal{S}_+^n, \mathfrak{g})$ wherein a $n$-dimensional SPD manifold $\mathcal{S}_+^n$ is endowed with a Log-Cholesky metric $\mathfrak{g}$ (Lin, 2019). The underlying principle of the Log-Cholesky metric lies in introducing a novel metric for the Cholesky manifold of lower triangular matrices with positive diagonal elements, denoted as $\mathcal{L}_+^n$, and subsequently projecting it onto the SPD manifold $\mathcal{S}_+^n$ through Cholesky decomposition. The

Cholesky decomposition represents $\boldsymbol{P} \in \mathcal{S}_+^n$ as the product of $\boldsymbol{L} \in \mathcal{L}_+^n$ and its transpose, i.e., $\boldsymbol{P} = \boldsymbol{L}\boldsymbol{L}^\top$. The smooth map between these two manifolds is defined as the Cholesky map $\mathscr{L} : \mathcal{S}_+^n \to \mathcal{L}_+^n$, and its inverse map is denoted as $\mathscr{S}$. Notably, these two maps establish a diffeomorphism between $\mathcal{L}_+^n$ and $\mathcal{S}_+^n$ (Lin, 2019). The differential $D_{\boldsymbol{L}}\mathscr{S} : \mathcal{T}_{\boldsymbol{L}}\mathcal{L}_+^n \to \mathcal{T}_{\mathscr{S}(\boldsymbol{L})}\mathcal{S}_+^n$ of map $\mathscr{S}$ at point $\boldsymbol{L}$ is given by

$$D_{\boldsymbol{L}}\mathscr{S} : \boldsymbol{Z} \mapsto \boldsymbol{L}\boldsymbol{Z}^\top + \boldsymbol{Z}\boldsymbol{L}^\top, \tag{2}$$

and the differential $D_{\boldsymbol{P}}\mathscr{L} : \mathcal{T}_{\boldsymbol{P}}\mathcal{S}_+^n \to \mathcal{T}_{\mathscr{L}(\boldsymbol{P})}\mathcal{L}_+^n$ of map $\mathscr{L}$ at point $\boldsymbol{P}$ is given by

$$D_{\boldsymbol{P}}\mathscr{L} : \boldsymbol{U} \mapsto \mathscr{L}(\boldsymbol{P}) \left( \mathscr{L}(\boldsymbol{P})^{-1} \boldsymbol{U} \mathscr{L}(\boldsymbol{P})^{-\top} \right)_{\frac{1}{2}}, \tag{3}$$

where $\boldsymbol{A}^{-\top 1}$ denotes the inverse of $\boldsymbol{A}^\top$ and $\boldsymbol{A}_{\frac{1}{2}}$ denotes the lower triangular part of $\boldsymbol{A}$ with the diagonal elements halved. With the diffeomorphism between $\mathcal{L}_+^n$ and $\mathcal{S}_+^n$, the Riemannian metric on $\mathcal{S}_+^n$ can be derived from $\mathcal{L}_+^n$. Given $\boldsymbol{Y}, \boldsymbol{Z} \in \mathcal{T}_{\boldsymbol{L}}\mathcal{L}_+^n$, Riemannian metric $\tilde{\mathfrak{g}}$ at $\boldsymbol{L} \in \mathcal{L}_+^n$ is defined as

$$\begin{aligned}
\tilde{\mathfrak{g}}_{\boldsymbol{L}}(\boldsymbol{Y}, \boldsymbol{Z}) &:= \langle \boldsymbol{Y}, \boldsymbol{Z} \rangle_{\boldsymbol{L}} \\
&= \langle \lfloor \boldsymbol{Y} \rfloor, \lfloor \boldsymbol{Z} \rfloor \rangle_F + \langle \mathbb{D}(\boldsymbol{L})^{-1}\mathbb{D}(\boldsymbol{Y}), \mathbb{D}(\boldsymbol{L})^{-1}\mathbb{D}(\boldsymbol{Z}) \rangle_F \\
&= \sum_{i>j} \boldsymbol{Y}_{ij}\boldsymbol{Z}_{ij} + \sum_{j=1}^n \boldsymbol{Y}_{jj}\boldsymbol{Z}_{jj}\boldsymbol{L}_{jj}^{-2},
\end{aligned} \tag{4}$$

where $\langle \cdot, \cdot \rangle_F$ denotes the Frobenius inner product and $\lfloor \boldsymbol{A} \rfloor$ denotes a $n \times n$ strict low-triangular matrix. For $i > j$, the elements in $(i, j)$ of $\lfloor \boldsymbol{A} \rfloor$ are represented by $\boldsymbol{A}_{ij}$; otherwise, the elements are set to 0. Moreover, $\mathbb{D}(\boldsymbol{A})$ denotes a $m \times m$ diagonal matrix, with the diagonal elements corresponding to $\boldsymbol{A}_{ii}$. Subsequently, we can transform the Riemannian metric from $\mathcal{L}_+^n$ to $\mathcal{S}_+^n$. Give $\boldsymbol{U}, \boldsymbol{V} \in \mathcal{T}_{\boldsymbol{P}}\mathcal{S}_+^n$, the **Riemannian metric** $\mathfrak{g}$ at $\boldsymbol{P} \in \mathcal{S}_+^n$ is defined as

$$\mathfrak{g}_{\boldsymbol{P}}(\boldsymbol{U}, \boldsymbol{V}) := \langle \boldsymbol{U}, \boldsymbol{V} \rangle_{\boldsymbol{P}} = \tilde{\mathfrak{g}}_{\mathscr{L}(\boldsymbol{P})}((D_{\boldsymbol{P}}\mathscr{L})(\boldsymbol{U}), (D_{\boldsymbol{P}}\mathscr{L})(\boldsymbol{V})). \tag{5}$$

**Geodesic distance** on a manifold generalizes the concept of straight lines in Euclidean geometry. The definition of geodesic distance holds significant importance in graph embedding, where the optimization objective frequently involves minimizing the geodesic distance between adjacent nodes. We formally define the geodesic distance $d_{\mathcal{S}_+^n}$ between points $\boldsymbol{P}, \boldsymbol{Q} \in \mathcal{S}_+^n$ through the geodesic distance $d_{\mathcal{L}_+^n}$ between points $\boldsymbol{L}, \boldsymbol{K} \in \mathcal{L}_+^n$, denoted as

$$d_{\mathcal{S}_+^n}(\boldsymbol{P}, \boldsymbol{Q}) = d_{\mathcal{L}_+^n}(\mathscr{L}(\boldsymbol{P}), \mathscr{L}(\boldsymbol{Q})), \tag{6}$$

$$d_{\mathcal{L}_+^n}(\boldsymbol{L}, \boldsymbol{K}) = (\|\lfloor \boldsymbol{L} \rfloor - \lfloor \boldsymbol{K} \rfloor\|_F^2 + \|\log \mathbb{D}(\boldsymbol{L}) - \log \mathbb{D}(\boldsymbol{K})\|_F^2)^{1/2}, \tag{7}$$

where $\|\boldsymbol{A}\|_F := \langle \boldsymbol{A}, \boldsymbol{A} \rangle_F^{1/2}$ denotes the Frobenius norm. The mapping between the tangent space $\mathcal{T}_{\boldsymbol{P}}\mathcal{S}_+^n$ and $\mathcal{S}_+^n$ undergoes a transformation via **Riemannian Exponential and Logarithmic maps**. We have the explicit expressions for the Riemannian exponential map on $\mathcal{S}_+^n$, which maps points on the SPD manifold $\mathcal{S}_+^n$ to the tangent space $\mathcal{T}_{\boldsymbol{P}}\mathcal{S}_+^n$ at $\boldsymbol{P} \in \mathcal{S}_+^n$ through the Riemannian exponential map on $\mathcal{L}_+^n$. The expression is given by

$$\mathrm{Exp}_{\boldsymbol{P}}(\boldsymbol{V}) = \left( \tilde{\mathrm{Exp}}_{\mathscr{L}(\boldsymbol{P})}((D_{\boldsymbol{P}}\mathscr{L})(\boldsymbol{V})) \right) \left( \tilde{\mathrm{Exp}}_{\mathscr{L}(\boldsymbol{P})}((D_{\boldsymbol{P}}\mathscr{L})(\boldsymbol{V})) \right)^\top, \tag{8}$$

$$\tilde{\mathrm{Exp}}_{\boldsymbol{L}}(\boldsymbol{Z}) = \lfloor \boldsymbol{L} \rfloor + \lfloor \boldsymbol{Z} \rfloor + \mathbb{D}(\boldsymbol{L}) \exp(\mathbb{D}(\boldsymbol{Z})\mathbb{D}(\boldsymbol{L})^{-1}). \tag{9}$$

Likewise, the Riemannian logarithmic map on $\mathcal{S}_+^n$ is also derived from the Riemannian logarithmic map on $\mathcal{L}_+^n$:

$$\mathrm{Log}_{\boldsymbol{P}}\boldsymbol{Q} = (D_{\mathscr{L}(\boldsymbol{P})}\mathscr{S})(\tilde{\mathrm{Log}}_{\mathscr{L}(\boldsymbol{P})}\mathscr{L}(\boldsymbol{Q})), \tag{10}$$

$$\tilde{\mathrm{Log}}_{\boldsymbol{L}}\boldsymbol{K} = \lfloor \boldsymbol{K} \rfloor - \lfloor \boldsymbol{L} \rfloor + \mathbb{D}(\boldsymbol{L}) \log(\mathbb{D}(\boldsymbol{L})^{-1}\mathbb{D}(\boldsymbol{K})). \tag{11}$$

**Rationale for the choice of the Log-Cholesky metric.** (1) **Aggregation analytics.** In the context of neighborhood aggregation of GNNs, the Fréchet mean of aggregated node neighborhoods serves as an efficient surrogate expression for information transmission, with its closed-form analytics being pivotal. When considering metrics on the SPD manifold, both Log-Cholesky metric (Lin,

---

[1]Concisely, we may abuse the symbol $\boldsymbol{A}$ to introduce the operations defined on matrices.

2019) and LEM (Arsigny et al., 2007) provide closed-form expressions for the Fréchet mean, while AIM (Pennec et al., 2006) necessitates incremental analytics for approximation. (2) **Computational tractability.** Despite LEM (Arsigny et al., 2007) possessing desirable properties such as Riemannian symmetry, geodesic completeness, and inverse consistency, the computation of Riemannian exponential and logarithmic maps for this metric involves evaluating an infinite series, incurring substantial computational costs. In summary, considering these factors, the Log-Cholesky metric is chosen due to its satisfaction of our requirements. This metric provides a closed-form expression for the Fréchet mean and computational efficiency, aligning well with the objectives of our research.

## 3 RIEMANNIAN GNNS WITH LOG-CHOLESKY METRIC

In this section, we reconstruct GNNs on the SPD manifold with the Log-Cholesky metric, encompassing feature transformation, neighborhood aggregation, non-linear activation, and multiclass logistic regression. Before this reconstruction process, fundamental components of GNNs are introduced through the classical Graph Convolutional Networks (GCN) model as an illustrative example.

### 3.1 BACKGROUND ON GRAPH NEURAL NETWORKS

**Problem setting.** Consider the graph $\mathcal{G} = (\mathcal{V}, \mathcal{E})$, where $\mathcal{V}$ denotes the node set and $\mathcal{E}$ denotes the edge set. Additionally, let $(\boldsymbol{x}_i^E)_{i \in \mathcal{V}}$ denote the $d$-dimensional node features in Euclidean space, denoted by the superscript $E$. Moreover, a subset of nodes $\mathcal{V}_t \subset \mathcal{V}$ is associated with class labels $\{y_i\}_{i \in \mathcal{V}_t}$. The objective of the *semi-supervised node classification task* is to predict the labels for nodes without labels $\{y_i\}_{i \in \mathcal{V} \setminus \mathcal{V}_t}$ or even for newly introduced nodes in the graph.

**Graph Convolutional Neural Networks (GCN).** Let $\mathcal{N}(i) = \{j : (i, j) \in \mathcal{E}\}$ denote the set of neighbors of node $i$, $\boldsymbol{W}^\ell$ be weights for layer $\ell$, and $\sigma(\cdot)$ denote a non-linear activation function. The general GCN message-passing scheme at layer $\ell$ for node $i$ consists of

$$\boldsymbol{h}_i^{\ell,E} = \boldsymbol{W}^\ell \boldsymbol{x}_i^{\ell-1,E}, \qquad \text{(Feature Transformation)}$$

$$\boldsymbol{x}_i^{\ell,E} = \sigma \left( \sum_{j \in \mathcal{N}(i) \cup i} w_{ij} \boldsymbol{h}_j^{\ell,E} \right), \qquad \text{(Neighborhood Aggregation)} \tag{12}$$

where aggregation weights $w_{i,j}$ can be computed using various mechanisms (Kipf & Welling, 2017; Veličković et al., 2018). Message passing is performed over multiple layers to iteratively propagate messages across neighborhoods.

Subsequently, we systematically elucidate the generalization of GNN components onto the SPD manifold. For brevity and without loss of generality, we eschew the use of superscripts to denote the number of layers and subscripts for node indices. Formally, we define the vector feature of a node in Euclidean space as $\boldsymbol{x}^E$, the matrix feature on $\mathcal{L}_+^n$ as $\boldsymbol{X}^{\mathcal{L}}$, and the matrix feature on $\mathcal{S}_+^n$ as $\boldsymbol{X}^{\mathcal{S}}$. Furthermore, the matrix feature following the application of each component is denoted as $\hat{\boldsymbol{X}}^{\mathcal{S}}$.

### 3.2 MAPPING FEATURES FROM EUCLIDEAN TO SPD MANIFOLD

For GNN operations, the initial mapping of each node's input feature $\boldsymbol{x}^E \in \mathbb{R}^d$ onto the SPD manifold $\mathcal{S}_+^n$ is essential. This involves a linear transformation, parameterized by $\boldsymbol{W}$, to map the input feature $\boldsymbol{x}^E$ onto the Euclidean space $\mathbb{R}^{n(n+1)/2}$. Subsequently, the Rectified Linear Unit (ReLU) activation function $\sigma$ is applied to ensure that the elements remain positive. The resulting feature is then reshaped into $n \times n$ lower triangular matrices, essentially formulated as a mapping $\varphi : \mathbb{R}^{n(n+1)/2} \to \mathcal{L}_+^n$:

$$\varphi : \boldsymbol{x} \mapsto \begin{pmatrix} x_1 & 0 & \cdots & 0 \\ x_2 & x_3 & \cdots & 0 \\ \vdots & \vdots & \ddots & \vdots \\ x_{n(n-1)/2+1} & x_{n(n-1)/2+2} & \cdots & x_{n(n+1)/2} \end{pmatrix}, \tag{13}$$

where $x_i$ denotes the $i$-th element of the vector $\boldsymbol{x}$. Finally, the Cholesky inverse map $\mathscr{S}$ is applied to map features from $\mathcal{L}_+^n$ onto $\mathcal{S}_+^n$. The initial mapping process can be formally summarized as

$$\boldsymbol{X}^{\mathcal{S}} = \mathscr{S}(\boldsymbol{X}^{\mathcal{L}}) = \boldsymbol{X}^{\mathcal{L}}(\boldsymbol{X}^{\mathcal{L}})^\top, \boldsymbol{X}^{\mathcal{L}} = \varphi(\sigma(\boldsymbol{W}\boldsymbol{x}^E)). \tag{14}$$

## 3.3 FEATURE TRANSFORMATION

In Euclidean GNNs, the feature transformation typically involves mapping node features from one layer's vector space to the subsequent layer's vector space. Unfortunately, the SPD manifold $\mathcal{S}_+^n$ lacks a well-defined algebraic structure akin to a vector space. Instead of resorting to Euclidean transformations on the tangent space $\mathcal{T}_{\boldsymbol{X}}\mathcal{S}_+^n$, we adopt a more fundamental approach achieving two simultaneous objectives: dimensional transformation and preservation of the manifold structure.

The prototypical symmetries of SPD manifold are parameterized by elements of the **General Linear (GL) Group** $GL(n; \mathbb{R})$: any invertible matrix $\boldsymbol{M} \in \mathbb{R}^{n \times n}$ defines a symmetry $\boldsymbol{P} \mapsto \boldsymbol{M}\boldsymbol{P}\boldsymbol{M}^\top$, operating on points $\boldsymbol{P} \in \mathcal{S}_+^n$, encompassing operations like translation, rotation, and reflection (López et al., 2021b). However, as an invertible squared matrix $\boldsymbol{M} \in \mathbb{R}^{n \times n}$ typically preserves the dimension of the matrix $\boldsymbol{P}$, we consider a more general concept of row full-rank matrices $\boldsymbol{M} \in \mathbb{R}^{p \times n}$, where $n \geq p$, ensuring the positive definiteness of the transformed matrices. Nonetheless, due to the non-compact nature of the space of row full-rank matrices, direct optimization becomes impractical (Huang & Van Gool, 2017). To address this, we further constrain the space to be orthogonal, yielding $\boldsymbol{M} \in SO(n) \setminus SO(n-p)$, where $SO(n)$ denotes the $n \times n$ **Special Orthogonal (SO) Group** (James, 1976). This space resides within the Stiefel manifold $St(p, n)$, where $n \geq p$. Consequently, we formally define the manifold-preserving feature transformation as

$$\hat{\boldsymbol{X}}^{\mathcal{S}} = \boldsymbol{M}\boldsymbol{X}^{\mathcal{S}}\boldsymbol{M}^\top, \boldsymbol{X}^{\mathcal{S}} \in \mathcal{S}_+^n, \hat{\boldsymbol{X}}^{\mathcal{S}} \in \mathcal{S}_+^p, \tag{15}$$

where $\boldsymbol{M} \in St(p, n)$ denotes the Riemannian parameters on the Stiefel manifold $St(p, n)$. This feature transformation enables the transformation of the feature $\boldsymbol{X}$ from $\mathcal{S}_+^n$ to $\mathcal{S}_+^p$.

## 3.4 NEIGHBORHOOD AGGREGATION

Neighborhood aggregation is a pivotal step in GNNs, serving to capture crucial neighborhood structures and features. Fundamentally, it calculates the Euclidean mean of the neighborhood features, a concept naturally extending to the Fréchet mean (Fréchet, 1948; Lou et al., 2020) for computing the centroid of neighborhood features on the Riemannian manifold. The core concept behind the Fréchet mean is to minimize an expectation of squared geodesic distances within a set of points $\boldsymbol{P}$, formulated as $F(\boldsymbol{C}) = \sum_{i=1}^m d_{\mathcal{M}}^2(\boldsymbol{C}, \boldsymbol{P}_i)$, where $\boldsymbol{C}$ denotes the expected centroid and $d_{\mathcal{M}}$ denotes the geodesic distance function on the manifold $\mathcal{M}$. The Log-Cholesky metric theoretically provides a closed and easily computable form for the Fréchet mean on the SPD manifold.

**Proposition 1.** *(Lin, 2019) Given a collection of points $\{\boldsymbol{P}_1, \cdots, \boldsymbol{P}_m\} \in \mathcal{S}_+^n$, the Fréchet mean on the SPD manifold $\mathcal{S}_+^n$ of these points, denoted by $\acute{\mathrm{F}}^{\mathcal{S}}(\cdot)$, can be formulated as*

$$\acute{\mathrm{F}}^{\mathcal{S}}(\boldsymbol{P}_1, \cdots, \boldsymbol{P}_m) = \acute{\mathrm{F}}^{\mathcal{L}}\big(\mathscr{L}(\boldsymbol{P}_1), \cdots, \mathscr{L}(\boldsymbol{P}_m)\big)\acute{\mathrm{F}}^{\mathcal{L}}\big(\mathscr{L}(\boldsymbol{P}_1), \cdots, \mathscr{L}(\boldsymbol{P}_m)\big)^\top, \tag{16}$$

$$\acute{\mathrm{F}}^{\mathcal{L}}(\boldsymbol{L}_1, \cdots, \boldsymbol{L}_m) = \frac{1}{m}\sum_{i=1}^m \lfloor \boldsymbol{L}_i \rfloor + \exp\big(\frac{1}{m}\sum_{i=1}^m \log \mathbb{D}(\boldsymbol{L}_i)\big). \tag{17}$$

Proposition 1 demonstrates that the closed-form Fréchet mean $\acute{\mathrm{F}}^{\mathcal{S}}$ of a collection of points on $\mathcal{S}_+^n$ can be derived from the mean representation $\acute{\mathrm{F}}^{\mathcal{L}}$ of the corresponding points on $\mathcal{L}_+^n$. With this proposition, we formally define the neighborhood aggregation of $\boldsymbol{X}^{\mathcal{S}} \in \mathcal{S}_+^n$ as

$$\hat{\boldsymbol{X}}^{\mathcal{S}} = \acute{\mathrm{F}}^{\mathcal{S}}(\mathcal{N}(\boldsymbol{X}^{\mathcal{S}})), \tag{18}$$

where $\mathcal{N}(\boldsymbol{X}^{\mathcal{S}})$ denotes the set of neighborhood representations of $\boldsymbol{X}^{\mathcal{S}} \in \mathcal{S}_+^n$.

## 3.5 NON-LINEAR ACTIVATION

In the realm of deep neural networks, various ReLUs have been proposed to enhance discriminative performance (Jarrett et al., 2009; Nair & Hinton, 2010). Consequently, it is crucial to incorporate

ReLU-like layers to introduce non-linearity into GNNs on the SPD manifold. Drawing inspiration from the concept of the point-wise non-linear activation $\max(0, x)$, we devise a non-linear function $\sigma^{\mathcal{S}}(\cdot)$ to rectify the SPD matrix $\boldsymbol{X}^{\mathcal{S}} \in \mathcal{S}_+^n$ by amplifying its small positive eigenvalues:

$$\hat{\boldsymbol{X}}^{\mathcal{S}} = \sigma^{\mathcal{S}}(\boldsymbol{X}^{\mathcal{S}}) = \left(\sigma^{\mathcal{L}}(\mathscr{L}(\boldsymbol{X}^{\mathcal{S}}))\right)\left(\sigma^{\mathcal{L}}(\mathscr{L}(\boldsymbol{X}^{\mathcal{S}}))\right)^{\top}, \tag{19}$$

$$\sigma^{\mathcal{L}}(\boldsymbol{X}^{\mathcal{L}}) = \lfloor \boldsymbol{X}^{\mathcal{L}} \rfloor + \max(\epsilon \boldsymbol{I}, \mathbb{D}(\boldsymbol{X}^{\mathcal{L}})), \tag{20}$$

where $\boldsymbol{I}$ denotes the identity matrix, and $\epsilon$ denotes the rectification threshold. It is essential to underline that the Cholesky decomposition $\boldsymbol{X}^{\mathcal{L}} = \mathscr{L}(\boldsymbol{X}^{\mathcal{S}})$ typically guarantees that the elements of $\mathbb{D}(\boldsymbol{X}^{\mathcal{L}})$ are inherently positive. Consequently, the ReLU operation $\sigma^{\mathcal{S}}(\cdot)$ defined on the SPD manifold does not inherently induce sparsity. In other words, by increasing the values of $\mathbb{D}(\boldsymbol{X}^{\mathcal{L}})$, our objective is to enhance the determinant of $\boldsymbol{X}^{\mathcal{S}}$ and thus elevate the eigenvalues collectively. Practically, the non-linear operation in Eqn. 20 can be integrated into Eqn. 17, thereby eliminating the computational overhead for a separate Cholesky decomposition.

### 3.6 MULTICLASS LOGISTIC REGRESSION ON SPD MANIFOLDS

In conventional GNNs, the routine utilization of linear classifiers critically hinges on the presupposition that data adheres to the principles of Euclidean geometry. However, if the data or features in question lack a Euclidean structure, the rationale for employing linear classifiers become less compelling, as highlighted by (Lebanon & Lafferty, 2004). Hyperplane classifiers present an appealing solution for harmonizing these two objectives: effectively fitting the training data while achieving improved geometric generalization.

Figure 2: A 3D example of the SPD hyperplane on $\mathcal{S}_+^2$. The gray convex conical point cloud delineates the $\mathcal{S}_+^2$ space. The green points, sampled within $\mathcal{S}_+^2$, outline the SPD hyperplane. Notably, this hyperplane is orthogonal to the red normal vector $\boldsymbol{V} \in \mathcal{T}_{\boldsymbol{P}}\mathcal{S}_+^2$ and passes through the point $\boldsymbol{P} \in \mathcal{S}_+^2$.

Given an input $\boldsymbol{x} \in \mathbb{R}^n$, the multiclass logistic regression (MLR) serves as an operation for predicting the probabilities of all target outcomes $k \in \{1, 2, \cdots, K\}$ for the objective variable $y$, expressed as follows:

$$p(y = k|\boldsymbol{x}) \propto \exp(v_k(\boldsymbol{x})), \tag{21}$$

where $v_k(x) = \langle \boldsymbol{a}_k, \boldsymbol{x} \rangle - b_k$, with $\boldsymbol{a}_k \in \mathbb{R}^n$ denoting a parameterized vector and $b_k \in \mathbb{R}$ denoting a scalar shift. As demonstrated in (Lebanon & Lafferty, 2004; Ganea et al., 2018), Euclidean MLR can be reformulated geometrically in terms of distances to hyperplanes:

$$p(y = k|\boldsymbol{x}) \propto \exp(\mathrm{sign}(\langle \boldsymbol{a}_k, \boldsymbol{x} \rangle - b_k)\|\boldsymbol{a}_k\| d(\boldsymbol{x}, H_{\boldsymbol{a}_k, b_k})), \tag{22}$$

where $H_{\boldsymbol{a}_k, b_k} = \{\boldsymbol{x} \in \mathbb{R}^n | \langle \boldsymbol{a}_k, \boldsymbol{x} \rangle - b_k = 0\}$ denotes the hyperplane parameterized by the normal vector $\boldsymbol{a}_k \in \mathbb{R}^n \setminus \{\boldsymbol{0}\}$ and the scalar shift $b_k \in \mathbb{R}$. Additionally, $d(\boldsymbol{x}, H_{\boldsymbol{a}_k, b_k})$ denotes the margin distance from the point $\boldsymbol{x}$ to the hyperplane $H_{\boldsymbol{a}_k, b_k}$. Theoretically, it is natural to generalize the Euclidean hyperplane to the SPD setting through the tangent space:

**Definition 1** (SPD Hyperplane). *Given $\boldsymbol{P} \in \mathcal{S}_+^n$, $\boldsymbol{V} \in \mathcal{T}_{\boldsymbol{P}}\mathcal{S}_+^n \setminus \{\boldsymbol{0}\}$, SPD hyperplane is defined as*

$$H_{\boldsymbol{V}, \boldsymbol{P}} = \{\boldsymbol{S} \in \mathcal{S}_+^n : \langle \mathrm{Log}_{\boldsymbol{P}}(\boldsymbol{S}), \boldsymbol{V} \rangle_{\boldsymbol{P}} = 0\}. \tag{23}$$

The hyperplane $H_{\boldsymbol{V}, \boldsymbol{P}}$ denotes the union of points $\boldsymbol{S} \in \mathcal{S}_+^n$ that their tangent vectors $\mathrm{Log}_{\boldsymbol{P}}(\boldsymbol{S}) \in \mathcal{T}_{\boldsymbol{P}}\mathcal{S}_+^n$ are orthogonal to the normal vector $\boldsymbol{V} \in \mathcal{T}_{\boldsymbol{P}}\mathcal{S}_+^n$ at point $\boldsymbol{P} \in \mathcal{S}_+^n$. Notice that our definition matches that of hypergyroplanes (Nguyen & Yang, 2023). A illustrative example of a 3D hyperplane on $\mathcal{S}_+^2$ is depicted in Fig. 2.

Subsequently, given an input $\boldsymbol{X} \in \mathcal{S}_+^n$, the generalization of MLR on the SPD manifold can be described as

$$p(y = k|\boldsymbol{X}) \propto \exp(\mathrm{sign}(\langle \boldsymbol{V}_k, \mathrm{Log}_{\boldsymbol{P}_k}(X) \rangle_{\boldsymbol{P}_k}) \|\boldsymbol{V}_k\|_{\boldsymbol{P}_k} d(\boldsymbol{X}, H_{\boldsymbol{V}_k, \boldsymbol{P}_k})), \tag{24}$$

where $\langle \cdot, \cdot \rangle_{\boldsymbol{P}_k}$ and $\| \cdot \|_{\boldsymbol{P}_k}$ denote the Riemannian inner product and norm at $\boldsymbol{P}_k$. For the calculation of $d(\boldsymbol{X}, H_{\boldsymbol{V}_k, \boldsymbol{P}_k})$, we provide Proposition 2 whose proof is deferred to Appendix A.1.

**Proposition 2.** *Given an input $\boldsymbol{X} \in \mathcal{S}_+^n$ and a hyperplane $H_{\boldsymbol{V},\boldsymbol{P}}$ parameterized by $\boldsymbol{P} \in \mathcal{S}_+^n$ and $\boldsymbol{V} \in \mathcal{T}_{\boldsymbol{P}} \mathcal{S}_+^n \setminus \{\boldsymbol{0}\}$ on $\mathcal{S}_+^n$, the margin distance from $\boldsymbol{X}$ to hyperplane $H_{\boldsymbol{V},\boldsymbol{P}}$ can be formulated as*

$$d(\boldsymbol{X}, H_{\boldsymbol{V},\boldsymbol{P}}) =$$
$$\frac{|\langle \lfloor \mathscr{L}(\boldsymbol{X}) \rfloor - \lfloor \mathscr{L}(\boldsymbol{P}) \rfloor + \log\left(\mathbb{D}(\mathscr{L}(\boldsymbol{P}))^{-1}\mathbb{D}(\mathscr{L}(\boldsymbol{X}))\right), \lfloor \tilde{\boldsymbol{V}} \rfloor + \mathbb{D}(\mathscr{L}(\boldsymbol{P}))^{-1}\mathbb{D}(\tilde{\boldsymbol{V}}) \rangle_F|}{\|\boldsymbol{V}\|_{\boldsymbol{P}}}, \quad (25)$$

*where $\tilde{\boldsymbol{V}} = \mathscr{L}(\boldsymbol{P})\left(\mathscr{L}(\boldsymbol{P})^{-1}\boldsymbol{V}\mathscr{L}(\boldsymbol{P})^{-\top}\right)_{\frac{1}{2}}$.*

In prior works (Lebanon & Lafferty, 2004; Ganea et al., 2018), MLR has been generalized to spherical and hyperbolic geometries. Here, we focus on the generalization of MLR to the SPD manifold, as established by Theorem 1, whose proof is provided in Appendix A.2.

**Theorem 1.** *Given a set of data $\{\boldsymbol{X} : \boldsymbol{X} \in \mathcal{S}_+^n\}$ that could be divided into $K$ classes s.t. $\{1, \cdots, K\}$ under MLR discrimination from the SPD manifold, let $\boldsymbol{V}_k \in \mathcal{T}_{\boldsymbol{P}_k} \mathcal{S}_+^n \setminus \{\boldsymbol{0}\}$ denote the normal vector of the hyperplane $H_{\boldsymbol{V}_k, \boldsymbol{P}_k}$, and $\boldsymbol{P}_k \in \mathcal{S}_+^n$ denote the hyperplane bias. The probability of $\boldsymbol{X}$ belonging to the class $k$ can be approximately defined as*

$$p(y = k | \boldsymbol{X}) \propto \exp(\langle \lfloor \mathscr{L}(\boldsymbol{X}) \rfloor - \lfloor \mathscr{L}(\boldsymbol{P}_k) \rfloor + \log\left(\mathbb{D}(\mathscr{L}(\boldsymbol{P}_k))^{-1}\mathbb{D}(\mathscr{L}(\boldsymbol{X}))\right),$$
$$\lfloor \tilde{\boldsymbol{V}}_k \rfloor + \mathbb{D}(\mathscr{L}(\boldsymbol{P}_k))^{-1}\mathbb{D}(\tilde{\boldsymbol{V}}_k) \rangle_F), \quad (26)$$

*where $\tilde{\boldsymbol{V}}_k = \mathscr{L}(\boldsymbol{P}_k)\left(\mathscr{L}(\boldsymbol{P}_k)^{-1}\boldsymbol{V}_k\mathscr{L}(\boldsymbol{P}_k)^{-\top}\right)_{\frac{1}{2}}$.*

Theorem 1 posits that the probability of node $\boldsymbol{X}$ belong to the class $k$ is proportional to the Riemannian inner product between the tangent vector of node $\boldsymbol{X}$ at $\boldsymbol{P}_k$ and the normal vector $\boldsymbol{V}_k$, which can be considered as a geometric generalization of the prototypical MLR presented in Eqn. 21.

## 4 EXPERIMENTS

In this section, we conduct experiments on a variety of real-world graphs to assess the efficacy of the proposed SPDGNN in the domain of semi-supervised node classification. We systematically compare the performance of SPDGNN against several Euclidean and hyperbolic GNNs. Additionally, we leverage the visualization of node embeddings and class hyperplanes to investigate the expressiveness of SPDGNN for modeling both Euclidean and hyperbolic geometries.

### 4.1 EXPERIMENTAL SETUP

**Datasets.** Our experiments employ four real-world datasets: `Disease`, `Airport`, `PubMed`, and `Cora`, recognized as benchmarks for the node classification task. These datasets have undergone preprocessing by Chami et al. (2019) and are accessible in their code repository[2]. Comprehensive statistics for these datasets are presented in Table 3, with $\delta$-hyperbolicity vales determined according to Chami et al. (2019). A lower $\delta$ signifies a more hyperbolic nature of the graph. Further, detailed information regarding the datasets and their splits can be found in Appendix C.1.

**Implementation details.** In our following experiments, we employ the standard GNN framework to learn the node representations, utilizing components defined on SPD manifold (refer to Section 3). Subsequently, these representations are fed to the MLR module (see Section 3.6), yielding probability for each class. Additionally, we incorporate the margin ranking loss, defined as $Loss = \max(p - \hat{p} + m, 0)$. In this expression, $p$ denotes the probability for the correct class, $\hat{p}$ denotes the probabilities for the incorrect classes, and $m$ is a non-negative margin hyper-parameter. Furthermore, we optimize our SPDGNN model using Riemannian Adam (Kochurov et al., 2020). The presented averages and standard deviations were derived from 10 independent runs. Detailed implementations are provided in Appendix C.2.

---

[2]https://github.com/HazyResearch/hgcn

Table 1: Evaluation results and comparison with competitive methods. F1 scores (%) are reported.

| | Dataset | Disease | Airport | PubMed | Cora |
|---|---|---|---|---|---|
| | Method | $\delta = 0$ | $\delta = 1$ | $\delta = 3.5$ | $\delta = 11$ |
| Euclidean | GCN | 69.7±0.4 | 81.4±0.6 | 78.1±0.2 | 81.3±0.3 |
| | GAT | 70.4±0.4 | 81.5±0.3 | 79.0±0.3 | 83.0±0.7 |
| | SAGE | 69.1±0.6 | 82.1±0.5 | 77.4±2.2 | 77.9±2.4 |
| | SGC | 69.5±0.2 | 80.6±0.1 | 78.9±0.0 | 81.0±0.1 |
| Hyperbolic | HGCN | 82.8±0.8 | 90.6±0.2 | 78.4±0.4 | 81.3±0.6 |
| | HAT | 83.6±0.9 | – | 78.6±0.5 | 83.1±0.6 |
| | LGCN | 84.4±0.8 | 90.9±1.7 | 78.6±0.7 | **83.3±0.7** |
| | HYPONET | 96.0±1.0 | 90.9±1.4 | 78.0±1.0 | 80.2±1.3 |
| | SPD4GCN | 91.1±3.5 | 65.8±3.4 | 78.1±0.6 | 80.2±1.4 |
| | SPDGNN | **96.9±0.9** | **94.9±1.3** | **79.3±0.7** | 80.5±3.2 |

**Competitors.** We compare our SPDGNN against three categories of methods: (1) Euclidean GNN models, namely GCN (Kipf & Welling, 2017), GAT (Veličković et al., 2018), SAGE (Hamilton et al., 2017), and SGC (Wu et al., 2019). (2) Hyperbolic GNNs, including HGCN (Chami et al., 2019), HAT (Zhang et al., 2021a), LGCN (Zhang et al., 2021b), HYPONET (Chen et al., 2022). (3) SPD4GCN (Zhao et al., 2023). To ensure fairness, we reproduce the outcomes of the first two categories as reported in (Chen et al., 2022). Meanwhile, the results of SPD4GCN[3] and our SPDGNN are obtained using the identical experimental settings and data split, aligning with the details in (Chami et al., 2019; Chen et al., 2022).

## 4.2 COMPARATIVE RESULTS

**Performance evaluation of SPDGNN against competitors.** Table 1 displays the experimental results, comparing between SPDGNN and other competitive models. Notably, SPDGNN consistently outperform all competitors on datasets with low hyperbolicity, underscoring the efficacy of SPDGNN in learning node representations adaptable to grid and hierarchical structures. Particularly noteworthy is SPDGNN's comprehensive superiority over SPD4GCN in terms of effectiveness. Moreover, we compare the training and inference times, as detailed in Table 5 within Appendix D. It is evident that SPDGNN is much faster than the SPD4GCN during both training and inference. Collectively, these results demonstrate the promising effectiveness and efficiency of SPDGNN.

Table 2: Ablation results of SPDGNN and its variants.

| Model | Disease | Airport | PubMed | Cora |
|---|---|---|---|---|
| **w/o Stiefel Linear** | 93.1±2.6 | 90.3±1.2 | 77.5±1.4 | 76.3±2.3 |
| **w/o Non-Linear** | 89.2±2.2 | 90.8±1.4 | 77.1±0.7 | 75.3±3.1 |
| **w/o SPD MLR** | 93.8±3.8 | 90.8±1.9 | 76.7±0.3 | 73.6±3.6 |
| **SPDGNN** | 96.9±0.9 | 94.9±1.3 | 79.3±0.7 | 80.5±3.2 |

**Enhancing node representations: evaluating component contributions.** We assess the impact of proposed components of GNNs on the SPD manifold by systematically examining variations derived from removing each component individually from SPDGNN. Specifically, we replace Stiefel linear (refer to Section 3.3) with the IsometryQR (López et al., 2021b; Zhao et al., 2023), directly omit non-linear layer (refer to Section 3.5), and replace the SPD MLR (refer to Section 3.6) with the Euclidean MLR. Results presented in Table 2 consistently reveal conspicuous performance degradation upon the removal of each component, thereby substantiating the effectiveness of the fully restructured SPDGNN. To thoroughly examine the effectiveness of SPD MLR, we showcase the 3D visual representations of class hyperplanes and node embeddings derived through Euclidean MLR and SPD MLR, as depicted in Fig. 3. Observing the illustration, it becomes apparent that, in contrast to the flat Euclidean hyperplanes portrayed in Figs 3(a), 3(c), the SPD hyperplanes in Figs 3(b), 3(d) encapsulate both flat and curved hyperplanes. These correspond, respectively, to the geometric structures of Euclidean and hyperbolic spaces. With these comprehensive hyperplanes, node embeddings

---

[3]https://github.com/andyweizhao/SPD4GNNs

from distinct catalogs are clearly separated. This substantiates the effectiveness of SPD MLR to simultaneously incorporate both Euclidean and hyperbolic geometries for classification. Additional perspectives of these hyperplanes are elucidated in Appendix D, providing a vivid illustration of the SPD manifold's rich geometric properties.

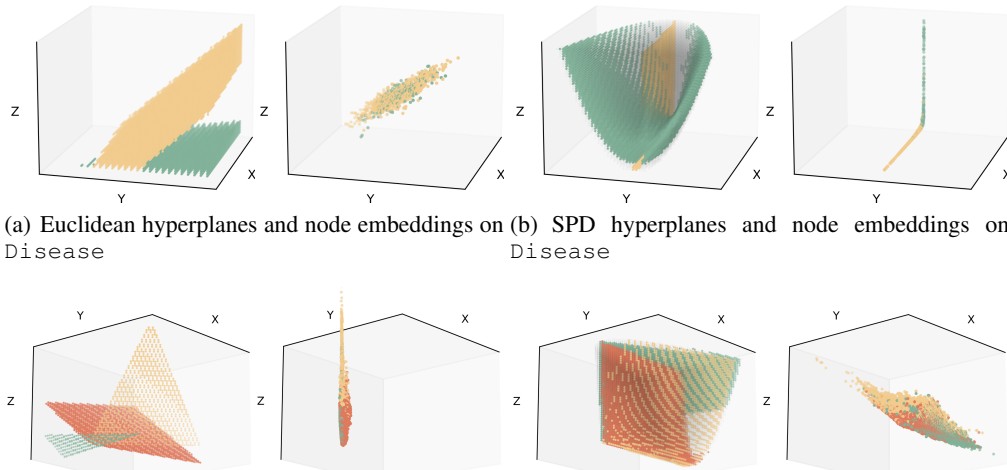

(a) Euclidean hyperplanes and node embeddings on `Disease`

(b) SPD hyperplanes and node embeddings on `Disease`

(c) Euclidean hyperplanes and node embeddings on `PubMed`

(d) SPD hyperplanes and node embeddings on `PubMed`

Figure 3: Direct Euclidean vs SPD MLR used to classify nodes on `Disease` and `PubMed`. In each subfigure, the left panel displays the class hyperplanes, while the right one exhibits the node embeddings. The node colors correspond to their respective catalogs. From a geometric perspective, subfigures (a) and (c) showcase Euclidean hyperplanes (depicting a flat observation) and node embeddings derived through Euclidean MLR. In contrast, subfigures (b) and (d) depict non-Euclidean hyperplanes (manifesting as curved surfaces) and node embeddings derived through SPD MLR.

### 4.3 ANALYSIS OF HYPER-PARAMETERS

**Impact of propagation depth on classification performance.** In Fig. 4 (left column), the outcomes of SPDGNN are depicted as the propagation steps range from 2 to 6. The results reveal that F1 scores reach optimal levels with shallow depths and gradually diminish as the number of propagation steps increases. This trend is attributed to the effectiveness of a moderate number of propagation steps in harnessing structural dependencies to enhance node representations. However, with an increase in propagation depth, there is a risk of over-smoothing, leading to the homogenization of node representations and making them indistinguishable across different categories.

**Impact of weight decay and margin on training.** Given the incorporation of multiple Riemannian parameters in SPDGNN, we study the performance nuances concerning weight decay and margin settings, as presented in Fig. 4 (middle and right columns, respectively). Our observations indicate that the performance of SPDGNN is notably dependent on the values assigned to weight decay and margin. Moreover, optimal values exhibit variability across different datasets.

## 5 CONCLUSIONS

In this study, we have reconstructed the formulation of graph neural networks (GNNs) on the symmetric positive definite (SPD) manifold, encompassing both Euclidean and hyperbolic entailments. Within this geometry-rich Riemannian structure, we have reformulated fundamental components, including linear transformation, neighborhood aggregation, non-linear activation, and multinomial logistic regression layers, incorporating novel geometric insights. Analysis and experimental results demonstrate the effectiveness and efficiency of the reconstructed SPDGNN in generating superior representations for both Euclidean and hyperbolic geometries, thereby enhancing the performance of the node classification task.

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

# A    DETAILED PROOFS

## A.1    PROOF FOR PROPOSITION 2

*Proof.* Obviously,

$$d(\boldsymbol{X}, H_{\boldsymbol{V}, \boldsymbol{P}}) := \inf_{\boldsymbol{Q} \in H_{\boldsymbol{V}, \boldsymbol{P}}} d_{\mathcal{S}_+^n}(\boldsymbol{X}, \boldsymbol{Q}). \tag{27}$$

Recall the geodesic distance in Eqn. 6

$$d_{\mathcal{S}_+^n}(\boldsymbol{X}, \boldsymbol{Q}) = d_{\mathcal{L}_+^n}(\mathcal{L}(\boldsymbol{X}), \mathcal{L}(\boldsymbol{Q})) \tag{28}$$

and the formula of distance between a point and a plane, we can rewrite Eqn. 27 as

$$d(\boldsymbol{X}, H_{\boldsymbol{V}, \boldsymbol{P}}) = d(\mathcal{L}(\boldsymbol{X}), H_{(D_{\boldsymbol{P}}\mathcal{L})(\boldsymbol{V}), \mathcal{L}(\boldsymbol{P})})$$

$$= \frac{|\langle \tilde{\mathrm{Log}}_{\mathcal{L}(\boldsymbol{P})}(\mathcal{L}(\boldsymbol{X})), (D_{\boldsymbol{P}}\mathcal{L})(\boldsymbol{V})\rangle_{\mathcal{L}(\boldsymbol{P})}|}{\|(D_{\boldsymbol{P}}\mathcal{L})(\boldsymbol{V})\|_{\mathcal{L}(\boldsymbol{P})}} \tag{29}$$

Concisely, we focus solely on deriving the numerator and omit the denominator and absolute value.

Based the Riemannian logarithmic map on $\mathcal{L}_+^n$ and the differential of Cholesky map $\mathcal{L}$ in Eqns. 11 and 3, respectively, we can expand the numerator in Eqn. 29 as

$$\langle \tilde{\mathrm{Log}}_{\mathcal{L}(\boldsymbol{P})}(\mathcal{L}(\boldsymbol{X})), (D_{\boldsymbol{P}}\mathcal{L})(\boldsymbol{V})\rangle_{\mathcal{L}(\boldsymbol{P})}$$

$$= \langle \lfloor \mathcal{L}(\boldsymbol{X}) \rfloor - \lfloor \mathcal{L}(\boldsymbol{P}) \rfloor + \mathbb{D}(\mathcal{L}(\boldsymbol{P})) \log \left(\mathbb{D}(\mathcal{L}(\boldsymbol{P}))^{-1} \mathbb{D}(\mathcal{L}(\boldsymbol{X}))\right),$$

$$\mathcal{L}(\boldsymbol{P}) \left(\mathcal{L}(\boldsymbol{P})^{-1} \boldsymbol{V} \mathcal{L}(\boldsymbol{P})^{-\top}\right)_{\frac{1}{2}} \rangle_{\mathcal{L}(\boldsymbol{P})}. \tag{30}$$

Concisely, let $\tilde{\boldsymbol{V}} := \mathcal{L}(\boldsymbol{P}) \left(\mathcal{L}(\boldsymbol{P})^{-1} \boldsymbol{V} \mathcal{L}(\boldsymbol{P})^{-\top}\right)_{\frac{1}{2}}$.

Based the Riemannian metric on $\mathcal{L}_+^n$ in Eqn. 4, we can expand Eqn. 30 as

$$\langle \lfloor \mathcal{L}(\boldsymbol{X}) \rfloor - \lfloor \mathcal{L}(\boldsymbol{P}) \rfloor + \mathbb{D}(\mathcal{L}(\boldsymbol{P})) \log \left(\mathbb{D}(\mathcal{L}(\boldsymbol{P}))^{-1} \mathbb{D}(\mathcal{L}(\boldsymbol{X}))\right), \tilde{\boldsymbol{V}}\rangle_{\mathcal{L}(\boldsymbol{P})}$$

$$= \langle \lfloor \mathcal{L}(\boldsymbol{X}) \rfloor - \lfloor \mathcal{L}(\boldsymbol{P}) \rfloor + \log \left(\mathbb{D}(\mathcal{L}(\boldsymbol{P}))^{-1} \mathbb{D}(\mathcal{L}(\boldsymbol{X}))\right), \lfloor \tilde{\boldsymbol{V}} \rfloor + \mathbb{D}(\mathcal{L}(\boldsymbol{P}))^{-1} \mathbb{D}(\tilde{\boldsymbol{V}})\rangle_F. \tag{31}$$

Besides, based the Riemannian metric on $\mathcal{S}_+^n$ in Eqn. 5, we can rewritten the denominator in Eqn. 29 as

$$|(D_{\boldsymbol{P}}\mathcal{L})(\boldsymbol{V})\|_{\mathcal{L}(\boldsymbol{P})} = \|\boldsymbol{V}\|_{\boldsymbol{P}}. \tag{32}$$

Thus, we can obtain

$$d(\boldsymbol{X}, H_{\boldsymbol{V}, \boldsymbol{P}}) =$$

$$\frac{|\langle \lfloor \mathcal{L}(\boldsymbol{X}) \rfloor - \lfloor \mathcal{L}(\boldsymbol{P}) \rfloor + \log \left(\mathbb{D}(\mathcal{L}(\boldsymbol{P}))^{-1} \mathbb{D}(\mathcal{L}(\boldsymbol{X}))\right), \lfloor \tilde{\boldsymbol{V}} \rfloor + \mathbb{D}(\mathcal{L}(\boldsymbol{P}))^{-1} \mathbb{D}(\tilde{\boldsymbol{V}})\rangle_F|}{\|\boldsymbol{V}\|_{\boldsymbol{P}}}, \tag{33}$$

where $\tilde{\boldsymbol{V}} = \mathcal{L}(\boldsymbol{P}) \left(\mathcal{L}(\boldsymbol{P})^{-1} \boldsymbol{V} \mathcal{L}(\boldsymbol{P})^{-\top}\right)_{\frac{1}{2}}$. $\quad\square$

A.2 Proof for Theorem 1

*Proof.* Recall the generalization of MLR on the SPD manifold in Eqn. 24

$$p(y = k|\boldsymbol{X}) \propto \exp(\text{sign}(\langle \boldsymbol{V}_k, \text{Log}_{\boldsymbol{P}_k}(X)\rangle_{\boldsymbol{P}_k}) \|\boldsymbol{V}_k\|_{\boldsymbol{P}_k} d(\boldsymbol{X}, H_{\boldsymbol{V}_k, \boldsymbol{P}_k})), \tag{34}$$

and the distance in Proposition 2

$$d(\boldsymbol{X}, H_{\boldsymbol{V}, \boldsymbol{P}}) =$$
$$\frac{|\langle \lfloor \mathscr{L}(\boldsymbol{X}) \rfloor - \lfloor \mathscr{L}(\boldsymbol{P}) \rfloor + \log\left(\mathbb{D}(\mathscr{L}(\boldsymbol{P}))^{-1}\mathbb{D}(\mathscr{L}(\boldsymbol{X}))\right), \lfloor \tilde{\boldsymbol{V}} \rfloor + \mathbb{D}(\mathscr{L}(\boldsymbol{P}))^{-1}\mathbb{D}(\tilde{\boldsymbol{V}})\rangle_F|}{\|\boldsymbol{V}\|_{\boldsymbol{P}}}, \tag{35}$$

where $\tilde{\boldsymbol{V}} = \mathscr{L}(\boldsymbol{P})\left(\mathscr{L}(\boldsymbol{P})^{-1}\boldsymbol{V}\mathscr{L}(\boldsymbol{P})^{-\top}\right)_{\frac{1}{2}}$.

Combining expression 34 and 35, we thus have

$$p(y = k|\boldsymbol{X}) \propto \exp(\langle \lfloor \mathscr{L}(\boldsymbol{X}) \rfloor - \lfloor \mathscr{L}(\boldsymbol{P}_k) \rfloor + \log\left(\mathbb{D}(\mathscr{L}(\boldsymbol{P}_k))^{-1}\mathbb{D}(\mathscr{L}(\boldsymbol{X}))\right),$$
$$\lfloor \tilde{\boldsymbol{V}}_k \rfloor + \mathbb{D}(\mathscr{L}(\boldsymbol{P}_k))^{-1}\mathbb{D}(\tilde{\boldsymbol{V}}_k)\rangle_F), \tag{36}$$

where $\tilde{\boldsymbol{V}}_k = \mathscr{L}(\boldsymbol{P}_k)\left(\mathscr{L}(\boldsymbol{P}_k)^{-1}\boldsymbol{V}_k\mathscr{L}(\boldsymbol{P}_k)^{-\top}\right)_{\frac{1}{2}}$. □

# B Related Works

**Graph neural networks.** Graph neural networks have expanded the capabilities of deep neural networks in handling graph data (Zhang et al., 2020). Kipf & Welling (2017) made groundbreaking contributions by introducing GCN, enabling the learning of node representations through iterative feature propagation on the graph structure. Subsequent to this pioneering work, numerous GNN methods have been proposed for diverse tasks, including node classification (Veličković et al., 2018; Wu et al., 2019; Gasteiger et al., 2018; Pei et al., 2019), link prediction (Zhang & Chen, 2018), and graph classification (Xu et al., 2018; Errica et al., 2019; Wijesinghe & Wang, 2021; Choi et al., 2023). However, these methods typically embed nodes into a Euclidean space, resulting in significant distortion when applied to real-world graphs exhibiting scale-free or hierarchical structure (Peng et al., 2021). In a non-Euclidean setting, hyperbolic space has gained increasing popularity for processing tree-like graph data. Ganea et al. (2018) presented a generalization of deep neural networks in hyperbolic space, specifically deriving hyperbolic multiclass logistic regression. Chami et al. (2019) proposed HGCN, performing network operations in the tangent space of the hyperbolic manifold. Zhang et al. (2021a) introduced a graph attention network in the Poincaré ball model to embed hierarchical and scale-free graphs with low distortion. Additional, Zhang et al. (2021b) proposed a neighborhood aggregation method based on the centroid of Lorentzian distance. Building upon this, Chen et al. (2022) further adapted the Lorentz transformations, incorporating boost and rotation operations, to formalize fully essential transformation in hyperbolic space. Moreover, HIE (Yang et al., 2023) leveraged hierarchical information inferred from the hyperbolic distance of each node to the origin, thereby enhancing current hyperbolic methods.

**SPD neural networks.** To take advantage of geometric deep learning techniques, considerable efforts have been devoted to generalizing Euclidean deep leaning to the realm of Riemannian geometry (Zhen et al., 2019; Chakraborty et al., 2020; Nguyen, 2021; Bronstein et al., 2021). Huang & Van Gool (2017); Wang et al. (2021) devised a densely connected feed-forward network explicitly tailored for the SPD manifold, incorporating a bi-linear mapping layer and a non-linear activation function. Expanding on this, Zhang et al. (2020) presented an SPD transformation network for action recognition, encompassing SPD convolutional, non-linear, and recursive layers. In the pursuit of enhancing expressiveness and interpretability of graph embeddings, López et al. (2021b) developed vector-valued distance and gyrovector calculus on the SPD manifold. Building upon this work, Zhao et al. (2023) conducted a preliminary exploration of implementing GNNs in the tangent space of the SPD manifold. Furthermore, Nguyen & Yang (2023) applied the theory of gyrogroups and gyrovector spaces to the study of matrix manifolds, successfully constructing neural networks on these manifolds.

Table 3: Datasets statistics.

| Dataset | #Nodes | #Features | #Classes | #Edges | $\delta$-hyperbolicity |
|---|---|---|---|---|---|
| Disease | 1,044 | 1,000 | 2 | 1,043 | 0 |
| Airport | 3,188 | 11 | 4 | 18,631 | 1 |
| PubMed | 19,717 | 500 | 3 | 88,651 | 3.5 |
| Cora | 2,708 | 1433 | 7 | 5,429 | 11 |

## C  Experimental Details

### C.1  Dataset Information

As outlined in Section 4.1, datasets employed in our experiments are all public available and widely used as standard benchmarks for evaluating graph representation learning tasks. The detailed information is provided below.

Disease (Anderson & May, 1991) is constructed based on the SIR disease spreading model. The labels indicate whether a node was infected or not, while the node features represent the susceptibility to the disease. This dataset comprises tree networks with 1,044 nodes, 1,043 edges, 1,000 features, and 2 classes.

Airport (Chami et al., 2019) is a flight network where nodes correspond to airports and edges represent airline routes. The node features contains geographic information and GDP of the country to which the airport belongs. The node labels indicate the population of the corresponding country. The flight network in this dataset consists of 3,188 nodes, 18,631 edges, 11 features, and 4 classes.

Cora (Sen et al., 2008) and PubMed (Szklarczyk et al., 2016) serve as standard benchmarks for citation networks. In these networks, nodes represent published scientific papers, and edges indicate the citation relationships between them. The labels assigned to the nodes pertain to academic topics. Specifically, Cora contains 2,708 nodes, 5,429 edges, 1,433 features and 7 classes, while PubMed consists of 19,717 nodes, 88,651 edges, 500 features and 3 classes.

**Data split.**  We split the nodes in Disease dataset into 30%, 10%, and 60%, and the nodes in Airport dataset into 70%, 15%, and 15%. For Cora and PubMed datasets, we used 20 labeled examples per class for training. The above splits are the same as those used in (Chami et al., 2019; Chen et al., 2022).

### C.2  Detailed Implementations

**GNN framework.**  As outlined in Section 3, our objective is to reconstruct the GNNs on the SPD manifold, encompassing feature transformation, neighborhood aggregation, non-linear activation, and MLR. By incorporating these reconstructed components into the canonical GNN architecture, we propose a novel model, referred to as SPDGNN, which operates fully on the SPD manifold. The pseudo code for the training process of SPDGNN is provided in Algorithm 1.

**Hyper-parameters.**  For experimental model, the hyper-parameters are tuned from the following search space: learning rate in $\{0.001, 0.005, 0.01\}$, dropout in $\{0, 0.3, 0.5, 0.7, 0.9\}$, weight decay in $\{0, 0.0001, 0.001, 0.01, 0.1\}$, dimension in $\{5, 10\}$, number of layer in $\{2, 3, 4, 5\}$, and margin in $\{0.25, 0.5, 1, 2, 5\}$. The optimal hyper-parameters on various datasets are summarized in Table 4.

**Training details.**  The SPDGNN model is trained with the semi-supervised classification loss on labeled training nodes. To determine the epoch selection, we employ the loss on the validation set as an indicator and report the testing results accordingly. During each run, the model is trained for a maximum of 2000 epochs, with early stopping applied after 500 epochs of no improvement. The testing result reported is obtained from the epoch with the lowest validation loss.

**Numerical stability.**  The effectiveness of the SPD manifold for geometric representation learning has been validated. However, this desirable property comes with a drawback: numerical instability leading to the failure of Cholesky decomposition. These instabilities are caused by imprecise

---

**Algorithm 1:** Training process of SPDGNN

---

**Input:** Graph $\mathcal{G}$, feature $\boldsymbol{x}$ for each node, label $\hat{y}$ for a fraction of nodes, propagation step $\ell$, and maximal epochs $T$.

Set $t = 0$.

Initialize parameters.

**while** $t \leq T$ and not converge **do**
    (**Feature Map**) $\boldsymbol{X}^{\mathcal{S}} = \mathscr{S}(\varphi(\sigma(\boldsymbol{W}\boldsymbol{x}^E)))$; (Eqn.14)

    **for** $j = 1$ **to** $\ell$ **do**
        (**Feature Transformation**) $\boldsymbol{X}^{\mathcal{S}} \leftarrow \boldsymbol{M}\boldsymbol{X}^{\mathcal{S}}\boldsymbol{M}^{\top}$; (Eqn. 15)

        (**Neighborhood Aggregation**) $\boldsymbol{X}^{\mathcal{S}} \leftarrow \acute{\mathrm{F}}^{\mathcal{S}}(\mathcal{N}(\boldsymbol{X}^{\mathcal{S}}))$; (Eqn. 18)

        (**Non-linear Activation**) $\boldsymbol{X}^{\mathcal{S}} = \sigma^{\mathcal{S}}(\boldsymbol{X}^{\mathcal{S}})$; (Eqn. 19)
    **end**

    (**SPD MLR**) Calculate probability for each class $p(y = k | \boldsymbol{X}^{\mathcal{S}})$;(Eqn. 26)

    Calculate loss and optimize parameters $\boldsymbol{W}$, $\boldsymbol{M}$, and $\{\boldsymbol{P}_k, \boldsymbol{V}_k | k \in 1, \cdots, K\}$.
**end**

---

Table 4: Hyper-parameters for SPDGNN.

| Hyper-parameters | Disease | Airport | PubMed | Cora |
|---|---|---|---|---|
| Learning Rate | 0.001 | 0.01 | 0.01 | 0.005 |
| Dropout | 0 | 0 | 0.9 | 0.9 |
| Weight Decay | 0.0001 | 0.0001 | 0.001 | 0.001 |
| Dimension | 10 | 10 | 5 | 5 |
| Layer | 3 | 2 | 4 | 4 |
| Margin | 1 | 1 | 2 | 1 |

floating-point arithmetic systems. Such errors can impact the positive-definiteness of node embeddings, ultimately resulting in the failure of Cholesky decomposition. To mitigate this issue, we adopt *double precision floating-point format* during computation and *clamp eigenvalues* to ensure their positive values.

## D    EXPERIMENT EXTENSIONS

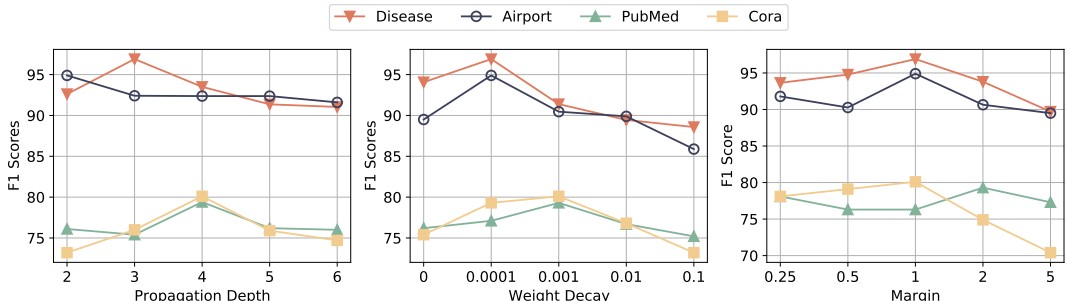

Figure 4: Hyper-parameter analysis for propagation depth, weight decay, and margin.

As the comprehensive rendition of Fig. 3, we magnificently showcase the hyperplanes learned by SPD MLR on `Disease`, `Airport`, and `PubMed` from multiple perspectives in Figs. 6-8, respectively, where $elev$ denotes the angle of elevation from the x-y plane and $zim$ denotes the angle of clockwise rotation around the z-axis.

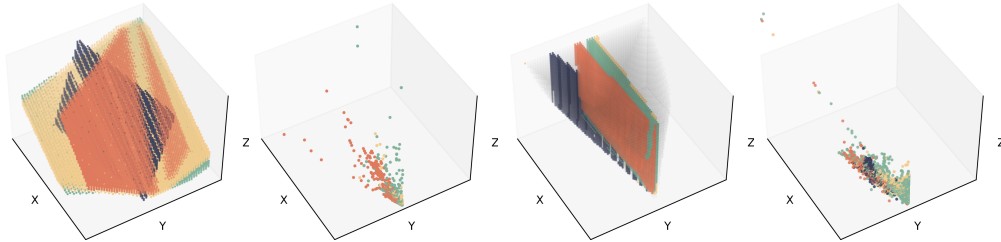

(a) Euclidean hyperplanes and node embeddings on Airport

(b) SPD hyperplanes and node embeddings on Airport

Figure 5: Direct Euclidean vs SPD MLR used to classify nodes on Airport.

Table 5: Comparison of training time per epoch and inference time on four datasets.

| Model | Disease | | Airport | | Pubmed | | Cora | |
|---|---|---|---|---|---|---|---|---|
| | TR(s) | IN(s) | TR(s) | IN(s) | TR(s) | IN(s) | TR(s) | IN(s) |
| HGCN | 0.011 | 0.010 | 0.014 | 0.005 | 0.018 | 0.006 | 0.011 | 0.007 |
| SPD4GCN | 1.068 | 1.144 | 3.115 | 3.265 | 18.002 | 18.479 | 2.779 | 2.702 |
| SPDGNN | 0.091 | 0.061 | 0.086 | 0.035 | 0.163 | 0.073 | 0.074 | 0.026 |

The SPD hyperplanes of $\mathcal{S}_+^2$ for Disease with z-axis variants are displayed in Fig. 6. By observing subplots (f) and (h), we can clearly see that the green hyperplane closely resembles the hyperboloid illustrated in Fig. 1, while the yellow hyperplane appears flat and exhibits mroe Euclidean geometry characteristics. This confirms the geometric manifold richness of the SPD manifold and further validate the effectiveness of the proposed components for geometric modeling on the SPD manifold.

The 3D visual representations of class hyperplanes and node embeddings derived through Euclidean MLR and SPD MLR, as depicted in Fig.5. The SPD hyperplanes of $\mathcal{S}_+^2$ for Airport with z-axis variants are displayed in Fig. 7. By examining subplots (b) and (g), we can observe that the navy-blue hyperplane exhibits a noticeable curvature, suggesting that this category may possess hierarchical structures with hyperbolicity. It should be note that, due to the inherent imprecision of floating-point arithmetic systems, slight distortions may occur on the hyperplanes. However, this does not compromise the characteristics of the hyperplanes.

The SPD hyperplanes of $\mathcal{S}_+^2$ for PubMed with z-axis variants are displayed in Fig. 8. By synthesizing the information from multiple subplots, we can unmistakably observe that the hyperplanes of the three categories exhibit a discernible hyperbolicity along convex cones. Additionally, their positioning is distinctive and corresponds closely to the node embeddings represented in Fig. 3.

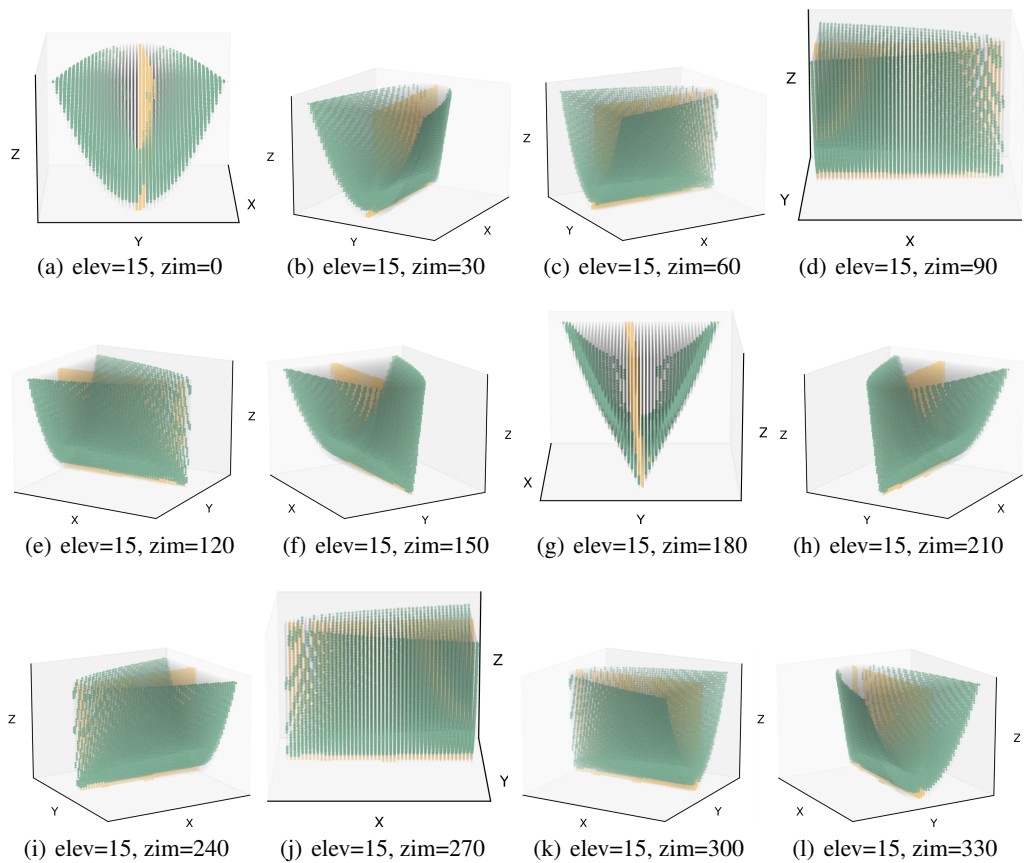

Figure 6: SPD hyperplanes of $\mathcal{S}_+^2$ for Disease with z-axis variants. Intuitively, SPD presents expressive structural hyperplanes for MLR including non-Euclidean curved surfaces.

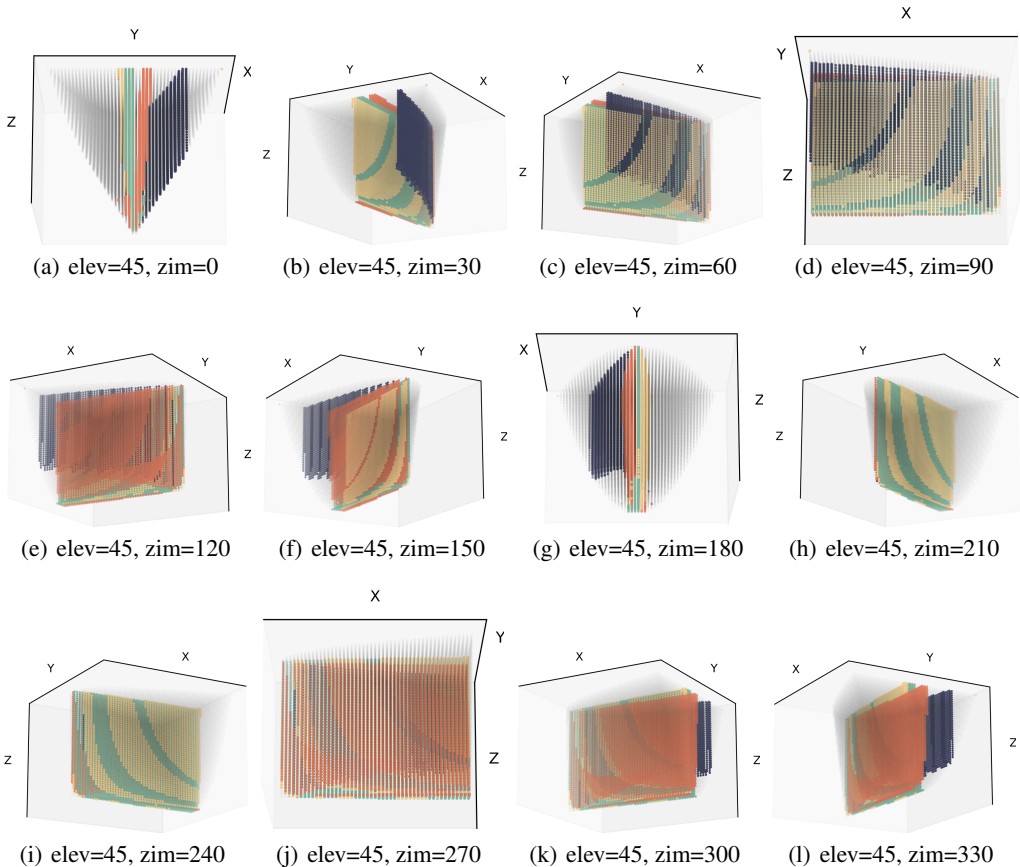

(a) elev=45, zim=0     (b) elev=45, zim=30     (c) elev=45, zim=60     (d) elev=45, zim=90

(e) elev=45, zim=120    (f) elev=45, zim=150    (g) elev=45, zim=180    (h) elev=45, zim=210

(i) elev=45, zim=240    (j) elev=45, zim=270    (k) elev=45, zim=300    (l) elev=45, zim=330

Figure 7: SPD hyperplanes of $\mathcal{S}_+^2$ for `Airport` with z-axis variants. Intuitively, SPD presents expressive structural hyperplanes for MLR including non-Euclidean curved surfaces.

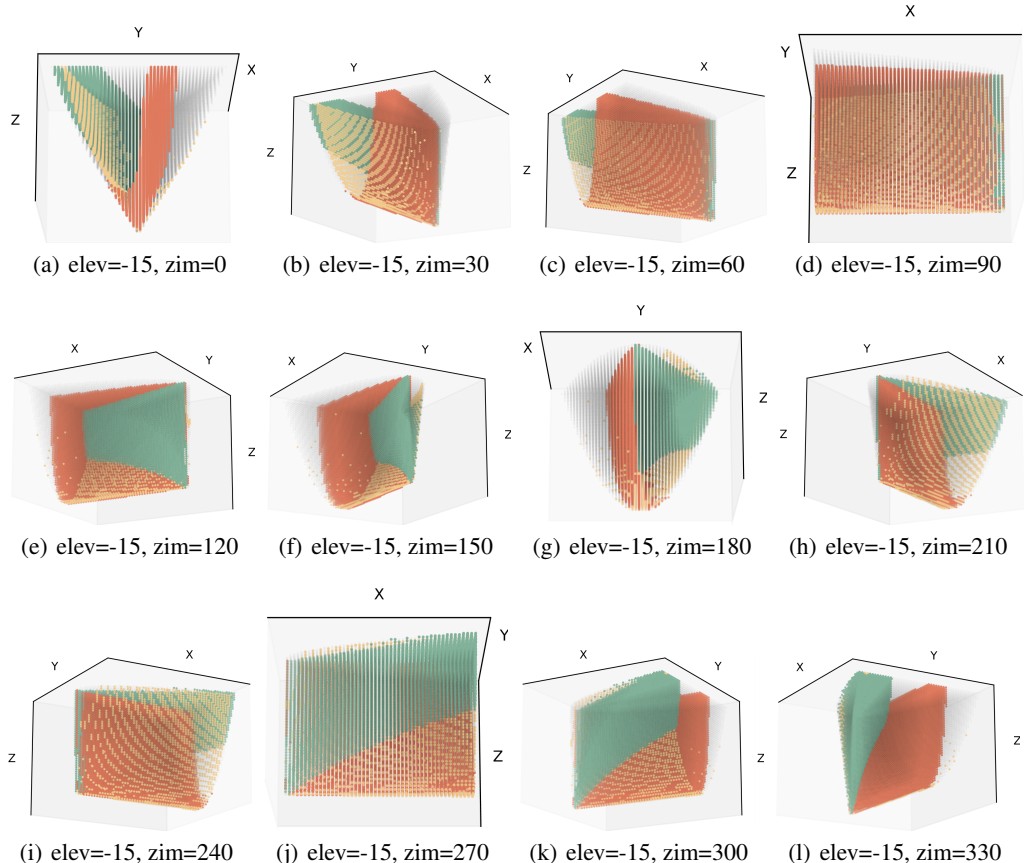

Figure 8: SPD hyperplanes of $\mathcal{S}_+^2$ for `PubMed` with z-axis variants. Intuitively, SPD presents expressive structural hyperplanes for MLR including non-Euclidean curved surfaces.

