# OpenReview forum: "Graph Neural Networks on Symmetric Positive Definite Manifold"
_ICLR.cc/2024/Conference — Submitted to ICLR 2024_

### Official Review · Reviewer_Jw38 · 2023-10-31

**Soundness:** 3 good
**Presentation:** 3 good
**Contribution:** 2 fair
**Rating:** 5
**Confidence:** 4

**Summary:**

The paper proposes a convolutional GNN whose features are represented as a symmetric positive definite matrix and measured by the log-Choleski metric. The Log-Choleski metric allows a closed-form expression of the Frechet mean, which is necessary for the aggregation step of the chosen GNN framework. Prior to this step, the feature transformation is performed by a double product with a learned parameter in the Stiefel manifold. Finally, the nonlinearity step is implemented as a rectification of the diagonal of the lower traingular part of a matrix, somehow modifying the eigenvalues of the matrix. The proposed SPDGNN architecture is compared with other geometric proposals for GNN, showing the performance of the model especially in the case of the airport database and disease.

**Strengths:**

- The architecture is novel and most of the choices of parametrization of each step are discussed, especially the choice of metrics.

- The ablation study and experiments on classical databases.

- the paper is written with particular attention to the notations necessary to capture the types of representation (SPD, lower triangular, ...)

**Weaknesses:**

- Missing at least one recent paper on the same topic with equivalent results on the same experiments (Hyperbolic Representation Learning : Revisiting and Advancing, M. Yang et al. ICML'23). Moreover, the numerical results for the methods presented in both papers are consistent, but not identical, with the globally higher performance reported in Yang et al.

- In the same vein, it would be good to discuss why geometric learning seems to underperform on PubMed and Cora and gives scores for other types of GNN on the same experiments (best competitor). A quick search gives scores on PubMed (91.4 points method from 2021) and Cora (90.16 points method from 2020) that are largely superior, with a caveat on the experimental setting.

- The case of edge-features is not addressed

**Questions:**

- Could a scaling factor be used in conjunction with the Stiefel parameter to give more freedom to the model?

- Is the initial mapping limited to a linear map, what is its influence?

- What is the expressive power of SPD-GNN?

- Any number of parameters gain compared to other hyperbolic and Riemannian GNNs?

---

> ### Author Response · Authors · 2023-11-20
> **Response to Reviewer Jw38 (Part 1 of 2)**
>
> Thank you for the thorough evaluation and constructive feedback on our paper. We appreciate the careful consideration of our proposed SPDGNN architecture. We have taken note of your comments and suggestions, and we are committed to addressing them in the revision. Below are our responses.
>
>
> > Q1: "**Missing at least one recent paper on the same topic....**"
>
>
> Thanks for sharing with us this up-to-date work. There are three main reasons for the slight discrepancies between its results and the ones we reported: 1) Variations in experimental settings, as it employs higher-dimensional representations (8, 64, 256 dimensions), whereas we chose matrix features of 5 and 10 dimensions, corresponding to node representations of 15 and 55 dimensions, respectively. 2) Differences in the reported metrics, as it presents accuracy for Cora, whereas our four datasets are uniformly reported in terms of F1-score. 3) Differences in datasets like Citeseer and PubMed . Furthermore, to our knowledge, its source code is currently not publicly available, making it challenging for us to consider it as our comparative method. Nonetheless, we will cite it and add relevant discussion in the revised version.
>
>
> > Q2: "**..., it would be good to discuss why geometric learning seems to underperform on PubMed and Cora and gives scores for other types of GNN on the same experiments...**"
>
> Thank you for your thoughtful comments. We understand your concern regarding the seemingly lower performance of geometric learning on PubMed and Cora. Allow us to provide a more detailed explanation:
> - **Problem Setting Clarification:**
> It's crucial to emphasize that the default development of GNNs is rooted in Euclidean space assumptions, and successful methods within this paradigm often presume settings in Euclidean space. However, these assumptions may not universally hold, prompting our exploration of alternative geometric frameworks, specifically hyperbolic and SPD geometries. Our endeavor surpasses the constraints of Euclidean geometry, delving into more intricate geometric attributes.
>
> - **Addressing Fundamental Problems:**
> Our approach is distinct in that it tackles a more fundamental problem within geometric learning. While state-of-the-art GNN methods developed in Euclidean space can be extended to hyperbolic or SPD manifolds, our work focuses on exploring the foundational geometric embedding space upon which GNNs rely. This emphasis on fundamental geometric properties allows us to understand the limitations and potentials of these spaces in the context of graph representation learning.
>
> - **Representative Baselines:**
> We exclusively compare our approach against representative methods endowed with geometric properties, specifically Euclidean and hyperbolic geometries. The comparison is intentionally limited to methods that **share a similar architectural foundation**, allowing for a more focused assessment of the inherent advantages and challenges associated with our proposed SPD geometry.
>
> In essence, our work extends beyond the conventional Euclidean assumptions of GNNs, exploring alternative geometric spaces to address fundamental challenges in graph representation learning. We acknowledge the potential superiority of state-of-the-art methods in Euclidean space, and we appreciate your suggestion to include additional context in our discussion. We will ensure that the revised manuscript provides a more comprehensive explanation of the motivation behind our geometric choices and their implications on the reported performance.
>
>
> > Q3: "**The case of edge-features is not addressed**"
>
> Thank you for your nice suggestion. Indeed, edge features constitute a pivotal concern. However, this manuscript concentrates on the exploration of latent geometric spaces, with further investigations into its higher echelons earmarked for our subsequent endeavors.
>
> > Q4: "**Could a scaling factor be used in conjunction with the Stiefel parameter to give more freedom to the model?**"
>
> Thank you for your suggestion, but preliminary analysis suggests that the inclusion of a scaling factor is unavailing for degrees of freedom. Stiefel parameters are acquired through Riemannian gradient descent, ensuring the positive definiteness of the transformed matrix. However, the utilizing of a scaling factor may potentially disrupt this assurance.
>
> >Q5: "**Is the initial mapping limited to a linear map, what is its influence?**"
>
> There is no absolute constraint. In truth, the objective of the initial mapping is to alter the dimensionality $n$ of input features to accommodate the matrix dimension $m$ (ensuring $n = m(m+1)/2$). The utilization of a linear mapping is the most direct and efficient means for this purpose.

---

> > ### Author Response · Authors · 2023-11-20
> > **Response to Reviewer Jw38 (Part 2 of 2)**
> >
> > > Q6: "**What is the expressive power of SPD-GNN?**"
> >
> > To avert unnecessary misconceptions, it is imperative to elucidate that, within the context of this manuscript, "expressiveness" pertains explicitly to the representation in geometric embedding and does not delve into an exploration of the expressive power of GNNs in addressing the graph isomorphism problem. Although the latter serves as the default interpretation of expressive power within the GNN context.
> >
> > > Q7: "**Any number of parameters gain compared to other hyperbolic and Riemannian GNNs?**"
> >
> > Thank you for your query. We have intentionally avoided introducing additional parameters in our model by aligning the expansion dimension of the matrix with the dimension of the corresponding vector. Specifically, the lower triangular elements of an $m \times m$ SPD matrix can be elegantly unfolded into a vector of dimension $m(m+1)/2$. In our experimental configuration, we have chosen a $5 \times 5$ SPD matrix, which corresponding to a $15-$dimensional vector representation. This setting ensures that the matrix representation of nodes incurs no additional parameters.

---

> > > ### Comment · Reviewer_Jw38 · 2023-11-23
> > >
> > > Thanks for the rebuttal.
> > > I still think the paper needs to be revised for acceptance, especially in terms of experimental settings and comparisons.

---

### Official Review · Reviewer_xhi5 · 2023-10-31

**Soundness:** 1 poor
**Presentation:** 1 poor
**Contribution:** 2 fair
**Rating:** 3
**Confidence:** 3

**Summary:**

This article introduces a new GNN architecture based on the idea that nodes are represented not as vectors but as symmetric positive-definite (SPD) matrices. The authors propose an architecture based on the Riemannian manifold associated with the Log-Cholesky metric. They apply this framework to semi-supervised node classification by defining logistic regression on the SPD manifold.

**Strengths:**

- The idea of extending embeddings to SPD matrices seems interesting.
- Utilizing the Riemannian framework with Log-Cholesky and logistic regressions on the SPD manifold appears to be a promising approach.

**Weaknesses:**

- Many of the contributions in this work are, in fact, already present in the literature, but the discussion regarding prior work is insufficient. The feature transformation with Stiefel matrices and the non-linear activation with the "ReLU-like" layer have already been proposed in [3]. However, at the point of defining these layers, the article does not clearly cite [3] as a reference but rather kind of presents them as original contributions. Additionally, the Riemannian/Cholesky approach is directly borrowed from [2], again without clear referencing when introduced. Moreover, there are significant interactions between the approach proposed here and that of [1], which is only discussed in the appendix. It is crucial to compare with [1] in the experiments and position this method in relation to it, as both seek GNNs where nodes are embedded in the space of SPD matrices. Consequently, it becomes challenging to precisely distinguish what constitutes contributions or ideas borrowed from other articles.

- One crucial point that doesn't seem to be discussed is the algorithmic complexity compared to standard approaches. From a memory perspective, the approach seems already highly expensive since each node is represented as a matrix with roughly $n^2$ parameters. Furthermore, in terms of algorithmic complexity, just the aggregation layer requires performing approximately $m$ Cholesky decompositions and computing an outer product with each of the representations (equation 12). So I doubt that this architecture is really applicable.

- I find that the article is generally quite confusing and not very clear. There are numerous and often ambiguous notations (for example, the notations for the Cholesky map and its inverse are almost identical), making it challenging to read. Additionally, the writing is quite heavy, with many vague and non-rigorous statements that don't convey a clear meaning. Here are a few examples:
  - "many complex graph data exhibit a profound non-Euclidean potential for analysis" (what is a "profound non-Euclidean potential"?)
  - "the SDP manifold [...] captures the hierarchical structure of datasets in hyperbolic subspaces while retaining Euclidean characteristics."
  - "In typical GNNs, a fundamental assumption is implicitly linked to linear classifiers, as they heavily rely on the Euclidean geometry of $\mathbb{R}^n$" (this is not true; you can have Euclidean geometry but use non-linear classifiers).

- The experimental setting in the article appears to lack clarity. The reported results include variances, but it's unclear on how many train/test splits these are based, neither if there is any cross-validation. Additionally, it's not specified whether the competing models were retrained for these experiments. If they were retrained, there is no information on how the retraining was conducted. Alternatively, it's unclear whether the performances were directly taken from the original articles. More transparency and details regarding the experimental setup and data handling would be beneficial for readers trying to understand and reproduce the results.

- References:

[1] Modeling Graphs Beyond Hyperbolic: Graph Neural Networks in Symmetric Positive Definite Matrices. Wei Zhao, Federico Lopez, J. Maxwell Riestenberg, Michael Strube, Diaaeldin Taha, Steve Trettel. ECML 2023.

[2] Geomnet: A neural network based on riemannian geometries of spd matrix space and cholesky space for 3d skeleton-based interaction recognition. Xuan Son Nguyen.

[3] A Riemannian Network for SPD Matrix Learning. Zhiwu Huang, Luc Van Gool. AAAI 2017.

**Questions:**

The Figure 2 is not so clear; what is $V$ on the Figure ? I assume that the manifold is in green while the hyperplane is in darkgrey, but it is very hard to see this ‘‘hyperplane'' and to interpret it.

---

> ### Author Response · Authors · 2023-11-20
> **Response to Reviewer xhi5 (Part 1 of 3)**
>
> Thank you for your constructive feedback and valuable suggestions. In response, we have endeavored to address your concerns and provide clarifications.
>
> > Q1: "**Many of the contributions in this work are, in fact, already present in the literature...**"
>
> Thank you for providing insightful comments and pointing out areas for improvement in our manuscript. While it is true that some aspects of our work share common ground with prior research, we contend that our study brings a unique contribution through generalizing the fundamental components of GNNs onto SPD manifold for learning node embeddings. Nevertheless, have carefully addressed each of your concerns in the revised manuscript.
> - **Feature Transformation**:
> The feature transformation utilizing Stiefiel bilinear mapping [3] is indeed a classical methodology, as described in our related work. However, it is crucial to note that, within the domain of graph neural networks and graph embedding [1,4], Isometry maps for SPD matrices have been the prevailing methodology. Our contribution lies in introducing dimensionality transformations atop Isometry maps for crafting our feature transformation layer. We have conducted a comprehensive ablation study, including relevant comparisons, to substantiate the superiority of our approach over IsometryQR. We appreciate your suggestion to enhance references in the subsection on feature transformation, particularly by incorporating [3].
> - **Non-linearity**:
> Regarding non-linearity and the choice of elevating eigenvalues, our contribution lies in propounding a non-linear methodology grounded in the log-Cholesky metric. In contrast to methods based on eigenvalue decomposition [3], our approach demonstrates efficiency by seamlessly integrating into the neighborhood aggregation layer (Eq.17) within the SPDGNN framework. This integration avoids the computational overhead associated with standalone eigenvalue decomposition.
>
> - **Riemannian/Cholesky Approach**:
> We respectfully disagree with the interpretation that our Riemannian/Cholesky approach is borrowed from [2]. The Log-Cholesky metric, as a Riemannian metric on the SPD manifold, has found widespread application in various domains. [2] focuses on a distinct task related to Skeleton-Based Interaction Recognition and employs Cholesky space differently. In contrast, our methodology extends the operations of GNN onto the SPD manifold based on the log-Cholesky metric. Additionally, we have duly cited [2] in the related work pertaining to SPD neural networks.
>
> - **Comparison with [1]**:
> We have addressed your suggestion to compare our method with [1] in the experiments. In the revised version, we have included SPD4GCN [1] as a baseline and reported the results of performance and efficiency.
>
> |          | **Method** | **Disease** | **Airport** | **PubMed** | **Cora**   |
> |----------|-------------|-------------|-------------|------------|------------|
> | **Euclidean** | GCN   | 69.7$\pm$0.4 | 81.4$\pm$0.6 | 78.1$\pm$0.2 | 81.3$\pm$0.3 |
> |              | GAT   | 70.4$\pm$0.4 | 81.5$\pm$0.3 | 79.0$\pm$0.3 | 83.0$\pm$0.7 |
> |              | SAGE  | 69.1$\pm$0.6 | 82.1$\pm$0.5 | 77.4$\pm$2.2 | 77.9$\pm$2.4 |
> |              | SGC   | 69.5$\pm$0.2 | 80.6$\pm$0.1 | 78.9$\pm$0.0 | 81.0$\pm$0.1 |
> | **Hyperbolic** | HGCN  | 82.8$\pm$0.8 | 90.6$\pm$0.2 | 78.4$\pm$0.4 | 81.3$\pm$0.6 |
> |               | HAT   | 83.6$\pm$0.9 |      --      | 78.6$\pm$0.5 | 83.1$\pm$0.6 |
> |               | LGCN  | 84.4$\pm$0.8 | 90.9$\pm$1.7 | 78.6$\pm$0.7 | **83.3**$\pm$**0.7** |
> |               | HYPONET | 96.0$\pm$1.0 | 90.9$\pm$1.4 | 78.0$\pm$1.0 | 80.2$\pm$1.3 |
> |   **SPD**            | SPD4GCN | 91.1$\pm$3.5 | 65.8$\pm$3.4 | 78.1$\pm$0.6 | 80.2$\pm$1.4 |
> |               | SPDGNN | **96.9**$\pm$**0.9** | **94.9**$\pm$**1.3** | **79.3**$\pm$**0.7** | 80.5$\pm$3.2 |
>
> We can observe that the performance of our SPDGNN surpasses that of [1] significantly, owing to our more comprehensive architecture.
>
> |**Method** | **Disease**| | **Airport**| | **PubMed** | |**Cora**  | |
> |----------------|-------|-------|-------|-------|-------|-------|-------|-------|
> |                | TR(s) | IN(s) | TR(s) | IN(s) | TR(s) | IN(s) | TR(s) | IN(s) |
> | **HGCN**       | 0.011 | 0.010 | 0.014 | 0.005 | 0.018 | 0.006 | 0.011 | 0.007 |
> | **SPD4GCN**    | 1.068 | 1.144 | 3.115 | 3.265 | 18.002| 18.479| 2.779 | 2.702 |
> | **SPDGNN**     | 0.091 | 0.061 | 0.086 | 0.035 | 0.163 | 0.073 | 0.074 | 0.026 |
>
> We can observe a significant enhancement in the efficiency of SPDGNN compared to [1], attributed to our metric selection and component optimizations as discussed in Section 2. While current SPD methods incur a certain increase in time overhead compared to hyperbolic methods due to various factors, including Cholesky decomposition, Riemannian gradient descent, and hyperplane classifier, the further exploration of its efficiency is warranted by the rich geometric properties.

---

> > ### Author Response · Authors · 2023-11-20
> > **Response to Reviewer xhi5 (Part 2 of 3)**
> >
> > > Q2:  "**One crucial point that doesn't seem to be discussed is the algorithmic complexity compared to standard approaches...**"
> >
> > We appreciate your insightful comments. In response to your comments, we have made the following clarifications and adjustments:
> >
> > - **Memory Perspective**:
> > The utilization of Symmetric Positive Definite (SPD) matrices implies a symmetric matrix, wherein the lower triangular elements of an m×m matrix can be elegantly unfolded into a vector of dimension m(m+1)/2. In our experimental configuration, a 5×5 SPD matrix is mapped to a 15-dimensional representation following conventional methodologies. As a result, the matrix representation of nodes incurs no additional memory overhead.
> >
> > - **Algorithmic Complexity**:
> > In terms of algorithmic complexity, as detailed in Section 2 (Rationale for the Choice of the Log-Cholesky Metric), it is highlighted that the computation of Riemannian exponential and logarithmic maps, as employed in [1], necessitates the evaluation of an infinite series, incurring substantial computational costs. In contrast, our adoption of the log-Cholesky metric effectively bypasses these intricacies, although it introduces the computational overhead of Cholesky decomposition.
> >
> > To empirically assess the efficiency of our proposed SPDGNN relative to SPD4GCN, we conducted a comparison of training and inference times for each epoch. The results, as provided in Q1, unequivocally demonstrate that SPDGNN significantly outperforms SPD4GCN. This observation serves as a clear indication of the superior efficiency of our approach in both the training and inference phases, reinforcing its practical advantages in terms of computational performance.
> >
> >
> > > Q3: "**I find that the article is generally quite confusing and not very clear...**"
> >
> > We are sorry for causing some unnecessary difficulty for readers. We have carefully polished our draft. The notations for the Cholesky map and its inverse are followed the symbols as defined in the original article [5]. we will address the specific vague and non-rigorous statements as follows:
> >
> > - many complex graph data exhibit a profound non-Euclidean potential for analysis $\rightarrow$   many complex graph datasets manifest characteristics that surpass the purview of conventional Euclidean analysis.
> >
> > - the SDP manifold [...] captures the hierarchical structure of datasets in hyperbolic subspaces while retaining Euclidean characteristics.  $\rightarrow$  Embedding graphs into the SPD manifold offers significant advantages, as it allows for the concurrent modeling of hierarchical structures in hyperbolic subspaces and grid structures in Euclidean subspaces.
> >
> > - In typical GNNs, a fundamental assumption is implicitly linked to linear classifiers, as they heavily rely on the Euclidean geometry $\rightarrow$ In conventional GNNs, the routine utilization of linear classifiers critically hinges on the presupposition that data adheres to the principles of Euclidean geometry.  (The previous expression does cause ambiguity. What we seek to convey is that the default deployment of a linear classifier relies on the foundational assumption of Euclidean geometry. In the absence of the Euclidean geometric structure, the linear classifier lacks cogency, prompting our adoption of a geometric hyperplane classifier. )
> >
> > > Q4:  "**The experimental setting in the article appears to lack clarity...**"
> >
> > Thanks for your valuable suggestions. We have added the information of experimental settings in the revised version.
> >
> > - **Data splits.** Disease dataset into 30\%, 10\%, and 60\%, and the nodes in Airport dataset into 70\%, 15\%, and 15\%. For Cora and PubMed datasets, we used 20 labeled examples per class for training. The above splits are the same as those used in [6].
> >
> > - **Reported resutls.** The presented averages and standard deviations were derived from 10 independent runs.
> >
> > - **Performance reproduction.** To ensure fairness, we reproduce the outcomes of the first two categories as reported in [7]. Meanwhile, the results of SPD4GCN (https://github.com/andyweizhao/SPD4GNNs) and our SPDGNN are obtained using the consistent experimental settings and data split, aligning with the details in [6,7].
> >
> > > Q5: "**The Figure 2 is not so clear...**"
> >
> > We are sorry for causing difficult for readers. We have revised the annotation and caption of Fig.2.  A 3D example of the SPD hyperplane on $\mathcal{S}^2_+$. The gray convex conical point cloud delineates the $\mathcal{S}^2_+$ space. The green points, sampled within $\mathcal{S}^2_+$, outline the SPD hyperplane. Notably, this hyperplane is orthogonal to the red normal vector $\mathbf{V}  \in T_\mathbf{P} \mathcal{S}^2_+$ and passes through the point $\mathbf{P} \in \mathcal{S}^2_+$.

---

> > > ### Author Response · Authors · 2023-11-20
> > > **Response to Reviewer xhi5 (Part 3 of 3)**
> > >
> > > [1]Zhao et al., Modeling Graphs Beyond Hyperbolic: Graph Neural Networks in Symmetric Positive Definite Matrices, ECML PKDD 2023.
> > >
> > > [2]Nguyen, Geomnet: A neural network based on riemannian geometries of spd matrix space and cholesky space for 3d skeleton-based interaction recognition, ICCV 2021.
> > >
> > > [3]Huang et al., A riemannian network for spd matrix learning, AAAI 2017.
> > >
> > > [4]Lopez et al., Vector-valued distance and gyrocalculus on the space of symmetric positive definite matrices, NeurIPS 2021.
> > >
> > > [5]Lin, Riemannian geometry of symmetric positive definite matrices via cholesky decomposition. SIAM Journal on Matrix Analysis and Applications, 2019.
> > >
> > > [6]Chami et al., Hyperbolic Graph Convolutional Neural Networks,NeurIPS 2019.
> > >
> > > [7]Chen et al., Fully Hyperbolic Neural Networks, ACL 2022.

---

### Official Review · Reviewer_KteU · 2023-10-31

**Soundness:** 4 excellent
**Presentation:** 2 fair
**Contribution:** 4 excellent
**Rating:** 8
**Confidence:** 4

**Summary:**

This paper presents a novel approach for defining graph neural networks that learn representations on the SPD manifold. The authors motivate their choice of the SPD manifold by the fact it exhibits both Euclidean and Hyperbolic geometry.

Starting from the Log-Cholesky metric, the authors derive closed-form expressions for weight updates, neighborhood aggregation (by computing the Frechet mean), and most notably for MLR on the SPD manifold (similar to the very important contribution of Ganea, 2018 that showed how to implement MLR for the Poincare Ball model of hyperbolic geometry).

The authors also propose a new non-linearity for SPD neural networks.

Finally, the authors conduct experiments on common node classification datasets, showing large improvements over previous methods.

**Strengths:**

The paper presents strong theoretical results, in particular a closed form expression for MLR on the SPD manifold can have impact on other SPD neural network architectures. The exposition is easy to follow, and the mathematics appear sound. Experimental performance shows a significant improvement in some benchmarks.

**Weaknesses:**

Although the paper reads well, there are some areas of lower clarity, I recommended proofreading to improve the writing a bit.

The paper does not cite previous work on SPD neural networks, e.g., SPDNet, SymNet, Chakraborty et al., etc. although they bear resemblance in the choice, e.g. of bilinear layers or of a rectifying function that amplifies small eigenvalues.

The experimental evaluation could be improved: Cora and Pubmed are saturated and unchallenging benchmarks, a better choice would be to use some of the more recent OGB benchmarks that come with a standardized evaluation procedure.

**Questions:**

I might be missing something but why does the choice of n >= p ensure positive-definiteness of the transformed matrices? (page 5)

Can the constraint of orthogonal matrices be relaxed?

With different formulations of the feature transformations and rectifying units compared to previous work, it is unclear whether part of the improved performance comes from these design choices. In the ablation study, can the authors clarify what alternatives were used when "removing" the Stiefel linear layers and the non-linearities? Could the authors compare against existing formulations from the SPD learning literature?

---

> ### Author Response · Authors · 2023-11-20
> **Response to Reviewer KteU (Part 1 of 2)**
>
> Thank you for taking time to review our paper and for providing valuable feedback. We appreciate your acknowledgement of our robust theoretical results and the improvements in experimental performance. We have thoroughly considered your suggestions and have addressed the raised points.
>
> > Q1: "**Although the paper reads well, there are some areas of lower clarity, I recommended proofreading to improve the writing a bit.**"
>
> Thanks for your suggestions. We have conducted a thorough proofreading to enhance the overall writing quality of the paper.
>
> > Q2: "**The paper does not cite previous work on SPD neural networks,...**"
>
> Thank you for sharing with us these related works. [1] devises a densely connected feed-forward network explicitly tailored for the SPD manifold, incorporating a bi-linear mapping layer and a non-linear activation function. Meanwhile, [2] applies a bi-linear mapping layer and a non-linear activation function for image set classification. [3] focuses on coping with manifold-value images and presents the analogue of convolution operations for manifold-value data.
> While our work is related, the key differences are three-fold: 1) We concentrate on graph neural networks for non-structural graph data.  2) We investigate the generalization of GNN components to SPD manifold with the Log-Cholesky metric and propose a framework named SPDGNN, encompassing linear, non-linear, neighborhood aggregation, and MLR layers. 3) We conduct experiments on four real-world graphs and validate our method against various GNN models under both Euclidean and hyperbolic geometries.
> Nonetheless, we will cite these works and provide relevant discussion in the revised version.
>
>
>
> > Q3: "**..., a better choice would be to use some of the more recent OGB benchmarks...**"
>
> Thank you for your valuable suggestions. Currently, the benchmarks commonly used in hyperbolic graph neural networks are still Disease, Airport, PubMed, and Cora [4,5]. Moreover, addressing OGB benchmarks poses additional resource demands for hyperbolic models due to the intricate mapping between the Riemannian manifold and the tangent space. We may consider expanding our experiments to OGB benchmarks after enhancing the efficiency of hyperbolic and other Riemannian manifold models.
>
> > Q4: "**I might be missing something but why does the choice of n >= p ensure positive-definiteness of the transformed matrices? (page 5)**"
>
> There might be a misunderstanding. It is not the case that $n \geq p$ ensures the positive definiteness of the transformed matrix, but rather it is guaranteed by a full-rank row matrix. The terms $p$ and $n$ merely denote the numbers of rows and columns, respectively.
>
> > Q5: "**Can the constraint of orthogonal matrices be relaxed?**"
>
> Certainly. Given that our transformation matrix $\mathbf{M}$ has $p$ rows and $n$ columns, where $p \leq n$, the orthogonality constraint solely limits that the rows to be orthogonal vectors. Hence, in essence, it is a semi-orthogonal constraint.
>
> > Q6: "**... In the ablation study, can the authors clarify what alternatives were used...**"
>
> Thank you for your insightful feedback. In response to your suggestions, we have adjusted the settings for the ablation study, conducting new experiments on variants involving linear layer ablation to provide a more comprehensive understanding of our components. Specifically, we replaced the Stiefel Linear layer with IsometryQR, an isometric mapping approach [5,6]. Our Stiefel linear layer, built upon this method, incorporates functionality for dimension transformation, allowing us to obtain more compact node embeddings.The ablation results also demonstrate the efficacy of our method. Furthermore, in the context of non-linear ablation, we chose to directly omit the non-linear module. This decision stems from the fact that the non-linear module proposed in our approach, based on the Log-Cholesky metric, can be seamlessly integrated into the neighborhood aggregation module. This integration eliminates the separate overhead incurred by Cholesky decomposition, which is a fundamental reason for not directly introducing non-linear methods based on eigenvalue decomposition [1,2].
>
> | **Model**                | **Disease** | **Airport** | **PubMed** | **Cora** |
> |--------------------------|----------------------|----------------------|---------------------|-------------------|
> | **w/o Stiefel Linear**   | 93.1$\pm$2.6         | 90.3$\pm$1.2         | 77.5$\pm$1.4        | 76.3$\pm$2.3      |
> | **w/o Non-Linear**        | 89.2$\pm$2.2         | 90.8$\pm$1.4         | 77.1$\pm$0.7        | 75.3$\pm$3.1      |
> | **w/o SPD MLR**           | 93.8$\pm$3.8         | 90.8$\pm$1.9         | 76.7$\pm$0.3        | 73.6$\pm$3.6      |
> | **SPDGNN**               | 96.9$\pm$0.9         | 94.9$\pm$1.3         | 79.3$\pm$0.7        | 80.5$\pm$3.2      |

---

> > ### Author Response · Authors · 2023-11-20
> > **Response to Reviewer KteU (Part 2 of 2)**
> >
> > [1] Huang et al., A riemannian network for spd matrix learning, AAAI 2017.
> >
> > [2] Wang et al., SymNet: A Simple Symmetric Positive Definite Manifold Deep Learning Method for Image Set Classification. IEEE TNNLS 2021.
> >
> > [3] Chakraborty et al., Manifoldnet: A deep neural network for manifold-valued data with applications. IEEE TPAMI, 2020.
> >
> > [4] M. Yang et al., Hyperbolic Representation Learning : Revisiting and Advancing, ICML 2023.
> >
> > [5] Zhao et al., Modeling Graphs Beyond Hyperbolic: Graph Neural Networks in Symmetric Positive Definite Matrices, ECML PKDD 2023.
> >
> > [6]Lopez et al., Vector-valued distance and gyrocalculus on the space of symmetric positive definite matrices, NeurIPS 2021.

---

### Official Review · Reviewer_KpWj · 2023-11-05

**Soundness:** 2 fair
**Presentation:** 1 poor
**Contribution:** 2 fair
**Rating:** 3
**Confidence:** 3

**Summary:**

This paper proposes to build Graph Neural Networks using the underlying geometry of symmetric positive definite matrices. The main motivation is that instead of working on Euclidean embeddings of graph features, embedding them in more geometric spaces like the Manifold of Symmetric Positive Definite Matrices (SPD) yields improved performance and richer representations.

The authors build on the framework of Log-Cholesky metrics that allow for mapping between the space of lower triangular matrices (with positive diagonal elements) and the SPD manifold. This unique decomposition allows for deriving the main components of GNNs like Feature Transformation, Neighbourhood aggregation, and Non-linear activation using specific formulas. The most significant amongst these is neighbourhood aggregation which can be done using a computationally attractive Frechet mean on the manifold.

Various experiments are reported to demonstrate that they improve upon standard GNN baselines. An ablation study is also reported to show the efficacy of individual components.

**Strengths:**

- I found the main idea of this paper: using Log-cholesky metric to map between SPD and Positive lower triangular matrix manifold to be interesting. To that aid, the various components (especially the frechet mean reformulation of neighbourhood aggregation) looks reasonable and interesting
-  The baseline experiments show decent proof of concept.

**Weaknesses:**

- The exposition of this paper can be significantly improved. I feel a significant lack of overall quality in the structure and messaging of this paper. The abstract is particularly too verbose and unclear.  To this aid, Figures 2 and 3 could be annotated and captioned more clearly to convey the message of the experiment.
- Some important baselines appear to be lacking like Zhao 2023 and Lopez et al 2021. I am especially critical of the lack of comparison with Zhao et.al 2023 which seems to propose an identical formulation (i.e. GNNs using SPD manifold - but the specific components are different to this paper).
- I miss any comparisons on runtime or complexity with previous methods. Again, Zhao et al 2023 seems significant baseline to compare with and report.

**Questions:**

Overall I feel this paper is not yet in the form that can be accepted at ICLR. Despite some similar recent works, the main idea is interesting. However, a lack of comparison with these baselines, and below-par overall writing quality makes it hard to promote acceptance at this point.

---

> ### Author Response · Authors · 2023-11-20
> **Response to Reviewer KpWj (Part 1 of 2)**
>
> Thank you for the time and valuable comments. We are glad that you liked our methodology and experimental endeavors. We notice that your concerns lie in two major points: the evaluation compared to (Zhao et.al 2023) and the clarity of exposition. Below, we offer detailed responses to these points and have made corresponding modifications to enhance the revised version in accordance with your concerns.
>
> >Q1: "**The exposition of this paper can be significantly improved. ... Figures 2 and 3 could be annotated and captioned more clearly...**"
>
> We regret any inconvenience caused to the readers and have meticulously refined our manuscript, particularly focusing on the logic and expression in the abstract, as outlined in our General Response. Additionally, we appreciate your suggestions regarding the clarity of Figures 2 and 3.
> - In Figure 2, we present a 3D illustration of the SPD hyperplane, parameterized by the normal vector $\mathbf{V}$ and a bias point $\mathbf{P}$. The gray convex conical point cloud outlines the SPD space, while the green points, sampled within SPD space, delineate the SPD hyperplane.
> - In Figure 3, we present a comparative analysis of 3D illustrations, depicting hyperplanes and node embeddings learned through Euclidean and SPD MLR, respectively. Unlike the flat Euclidean hyperplanes, the SPD hyperplanes encompass both flat and curved hyperplanes, corresponding to the geometric structures of Euclidean and hyperbolic spaces, respectively. These comprehensive hyperplanes effectively discriminate node embeddings from distinct catalogs (colors), demonstrating the efficacy of SPD MLR in concurrently incorporating both Euclidean and hyperbolic geometries for classification.
>
> >Q2: "**Some important baselines appear to be lacking... & I miss any comparisons on runtime...**"
>
> Thank you for your valuable suggestions. Considering the scope of [1] does not align with the domain of graph neural networks, while [2] represents a direct extension for it, we exclusively choose SPD4GCN [2] as our comparative method. Noteworthy distinctions between our approach and [2] include:
>
> - **Metric**: Our proposed operations are grounded in the Log-Cholesky metric, whereas [2] relies on the Gyrocalculus from [1].
>
> - **Architecture**: In addition to the fundamental neighborhood aggregation layer in GNNs, we introduce dimension-varying linear transformation and SPD MLR layers, setting our approach apart from [2].
>
> We have meticulously reproduced the experimental results of [2] using their source code, adhering to consistent experimental settings and data partitioning akin to other baselines. Due to variations in experimental settings, there are certain discrepancies compared to the original results; however, the overall trends remain consistent, particularly in the case of suboptimal results observed on the Airport dataset. These results have been integrated into the revised version of the manuscript.
>
> |          | **Method** | **Disease** | **Airport** | **PubMed** | **Cora**   |
> |----------|-------------|-------------|-------------|------------|------------|
> | **Euclidean** | GCN   | 69.7$\pm$0.4 | 81.4$\pm$0.6 | 78.1$\pm$0.2 | 81.3$\pm$0.3 |
> |              | GAT   | 70.4$\pm$0.4 | 81.5$\pm$0.3 | 79.0$\pm$0.3 | 83.0$\pm$0.7 |
> |              | SAGE  | 69.1$\pm$0.6 | 82.1$\pm$0.5 | 77.4$\pm$2.2 | 77.9$\pm$2.4 |
> |              | SGC   | 69.5$\pm$0.2 | 80.6$\pm$0.1 | 78.9$\pm$0.0 | 81.0$\pm$0.1 |
> | **Hyperbolic** | HGCN  | 82.8$\pm$0.8 | 90.6$\pm$0.2 | 78.4$\pm$0.4 | 81.3$\pm$0.6 |
> |               | HAT   | 83.6$\pm$0.9 |      --      | 78.6$\pm$0.5 | 83.1$\pm$0.6 |
> |               | LGCN  | 84.4$\pm$0.8 | 90.9$\pm$1.7 | 78.6$\pm$0.7 | **83.3**$\pm$**0.7** |
> |               | HYPONET | 96.0$\pm$1.0 | 90.9$\pm$1.4 | 78.0$\pm$1.0 | 80.2$\pm$1.3 |
> |   **SPD**            | SPD4GCN | 91.1$\pm$3.5 | 65.8$\pm$3.4 | 78.1$\pm$0.6 | 80.2$\pm$1.4 |
> |               | SPDGNN | **96.9**$\pm$**0.9** | **94.9**$\pm$**1.3** | **79.3**$\pm$**0.7** | 80.5$\pm$3.2 |
>
> We can observe that the performance of our SPDGNN surpasses that of [2] significantly, owing to our more comprehensive architecture.
>
> |**Method** | **Disease**| | **Airport**| | **PubMed** | |**Cora**  | |
> |----------------|-------|-------|-------|-------|-------|-------|-------|-------|
> |                | TR(s) | IN(s) | TR(s) | IN(s) | TR(s) | IN(s) | TR(s) | IN(s) |
> | **HGCN**       | 0.011 | 0.010 | 0.014 | 0.005 | 0.018 | 0.006 | 0.011 | 0.007 |
> | **SPD4GCN**    | 1.068 | 1.144 | 3.115 | 3.265 | 18.002| 18.479| 2.779 | 2.702 |
> | **SPDGNN**     | 0.091 | 0.061 | 0.086 | 0.035 | 0.163 | 0.073 | 0.074 | 0.026 |

---

> > ### Author Response · Authors · 2023-11-20
> > **Response to Reviewer KpWj (Part 2 of 2)**
> >
> > We can observe a significant enhancement in the efficiency of SPDGNN compared to [2], attributed to our metric selection and component optimizations as discussed in Section 2. While current SPD methods incur a certain increase in time overhead compared to hyperbolic methods due to various factors, including Cholesky decomposition, Riemannian gradient descent, and hyperplane classifier, the further exploration of its efficiency is warranted by the rich geometric properties.
> >
> > [1] Lopez et al., Vector-valued distance and gyrocalculus on the space of symmetric positive definite matrices, NeurIPS 2021.
> >
> > [2] Zhao et al., Modeling Graphs Beyond Hyperbolic: Graph Neural Networks in Symmetric Positive Definite Matrices, ECML PKDD 2023.

---

### Author Response · Authors · 2023-11-20
**General Response by Authors**

Dear Area Chairs and Reviewers,

We appreciate the reviewers' time, valuable comments and constructive suggestions.
Overall, the reviewers recognized our reasonable methods (KpWj, KteU, xhi5, Jw38) and strong empirical results with extensive experiments (KpWj, KteU, xhi5, Jw38).
The major concerns lie in exposition of this paper (KpWj, KteU, xhi5), lack of an important baseline (Zhao et al., ECML PKDD 2023.) (KpWj, KteU, xhi5, Jw38), comparisons on runtime (KpWj, xhi5), and additional experimental details ask by KteU, KteU, Jw38.

In the revised version, we have carefully scrutinized ambiguous exposition, added the experimental results and runtime of the suggested baselines, and conducted a more comprehensive experimental analysis based on the results. To clarify upfront some potential misunderstandings that may influence how our work is interpreted, we first restate our contributions.

- **Problem and Motivation** (See our Section 1) Geometric deep learning endows graph neural networks (GNNs) with some symmetry aesthetics from the inherent principles of the underlying graph structures. However, conventional modeling in Euclidean or hyperbolic geometry, often presupposes specific geometric properties for graphs, thereby neglecting the intricate actual structures. To address this limitation, we generalize the foundational components of GNNs to the Symmetric Positive Definite (SPD) manifold. This manifold theoretically endowed with a rich geometric structure that encompasses both Euclidean and hyperbolic projection subspaces.

- **Methodology** (See our Section 3) We present a reconstruction of GNNs on the SPD manifold with manifold-preserving linear transformation, neighborhood aggregation, non-linear activation, and multinomial logistic regression. In this framework, the Log-Cholesky metric is employed to derive the closed-form Fréchet mean representation for neighborhood aggregation, ensuring computational tractability in learning geometric embeddings. Also, to our knowledge, we are the first to leverage SPD manifold with the Log-Cholesky metric for learning graph embeddings.

- **Evaluation** (See our Section 4) We conduct comprehensive experiments, including performance and runtime comparisons with methods operating in Euclidean, hyperbolic, and SPD spaces, to evaluate our method. The results indicate superior performance over existing GNN methods and a significant enhancement in efficiency compared to the SPD method.


We next provide detailed answers to all the specific questions raised by the reviewers. Further discussions are welcomed to facilitate the reviewing process towards a comprehensive evaluation of our work.

---

### Meta-Review · Area_Chair_3EXh · 2023-12-08

**Metareview:**

In this submission, the authors generalize GNNs on the SPD manifold, reconstructing key components of GNNs (e.g., linear transformation, message-passing, activations, and so on) with manifold structure-preserving properties. Specifically, the Log-Cholesky metric is applied to implement the closed-form Frechet mean when aggregating neighborhood information, which is interesting. Experiments demonstrate the potential of the proposed method to some extent.

Strengths: (a) The motivation of this work is clear and reasonable --- when implementing a GNN model, taking the geometrical property of the target graph into account is necessary. (b) All reviewers admit the novelty of the proposed method, especially the usage of the Log-Cholesky metric.

Weaknesses: (a) Three reviewers have concerns about the solidness of the experimental part. More deep geometric learning should be considered as baselines. (b) Additionally, the datasets used in the experiments are relatively simple. It is unknown whether the proposed method is applicable to heterophilic graphs and real-world large-scale graphs. (c) The writing and the organization of this submission should be enhanced.

**Justification For Why Not Higher Score:**

Although the topic is important and the technical route is novel and interesting, the experimental part of this submission is not solid, which is insufficient to demonstrate the superiority and feasibility of the proposed method in practice. Additionally, the writing and organization of this submission should be enhanced further.

**Justification For Why Not Lower Score:**

N/A

---

### Decision · Program_Chairs · 2024-01-16

Reject